# MVInpainter: Learning Multi-View Consistent Inpainting to Bridge 2D and 3D Editing

**Chenjie Cao**[1,2,3], **Chaohui Yu**[2,3], **Fan Wang**[2,3], **Xiangyang Xue**[1], **Yanwei Fu**[1*]

[1]Fudan University, [2]DAMO Academy, Alibaba Group, [3]Hupan Lab

{caochenjie.ccj,huakun.ych,fan.w}@alibaba-inc.com
{xyxue,yanweifu}@fudan.edu.cn

## Abstract

Novel View Synthesis (NVS) and 3D generation have recently achieved prominent improvements. However, these works mainly focus on confined categories or synthetic 3D assets, which are discouraged from generalizing to challenging in-the-wild scenes and fail to be employed with 2D synthesis directly. Moreover, these methods heavily depended on camera poses, limiting their real-world applications. To overcome these issues, we propose MVInpainter, re-formulating the 3D editing as a multi-view 2D inpainting task. Specifically, MVInpainter partially inpaints multi-view images with the reference guidance rather than intractably generating an entirely novel view from scratch, which largely simplifies the difficulty of in-the-wild NVS and leverages unmasked clues instead of explicit pose conditions. To ensure cross-view consistency, MVInpainter is enhanced by video priors from motion components and appearance guidance from concatenated reference key&value attention. Furthermore, MVInpainter incorporates slot attention to aggregate high-level optical flow features from unmasked regions to control the camera movement with pose-free training and inference. Sufficient scene-level experiments on both object-centric and forward-facing datasets verify the effectiveness of MVInpainter, including diverse tasks, such as multi-view object removal, synthesis, insertion, and replacement. The project page is https://ewrfcas.github.io/MVInpainter/.

## 1 Introduction

This paper studies editing 3D scenes by expanding one or few 2D manipulated references to other observed views. Particularly, with the development of diffusion-based text-to-image (T2I) models [59, 63, 4, 54], we have seen substantial success in novel view synthesis (NVS) [23, 30, 97], 3D generation [55, 40, 79, 96, 44, 67, 43, 66], and controllable generation [60, 98, 10, 14]. But most existing synthesis methods [14, 98, 10] have only been proven useful in 2D scenarios. It is intuitive to extend these pioneering methods to multi-view scenarios to bridge the gap between 2D and 3D editing. This raises the question: how to make a unified framework to generate vivid multi-view foreground objects seamlessly integrated with their surroundings?

Despite the achievements in NVS and 3D generation, achieving multi-view consistent scene editing by inserting, removing, replacing objects like Fig. 1 still present challenges. 1) *3D object generation struggles to generalize to scene-level editing.* Most object-centric 3D generation methods [44, 67, 31] are trained on large-scale synthetic datasets [18, 17, 31] with simplistic backgrounds, neglecting real-world factors like illumination and shadows, which are essential for natural editing. While some methods integrate predefined 3D assets into NeRFs [65] and 3D Gaussian Splatting (3DGS) [15], they still struggle with blending foreground and background elements seamlessly. 2) *NVS methods have*

---

*Corresponding Author.

38th Conference on Neural Information Processing Systems (NeurIPS 2024).

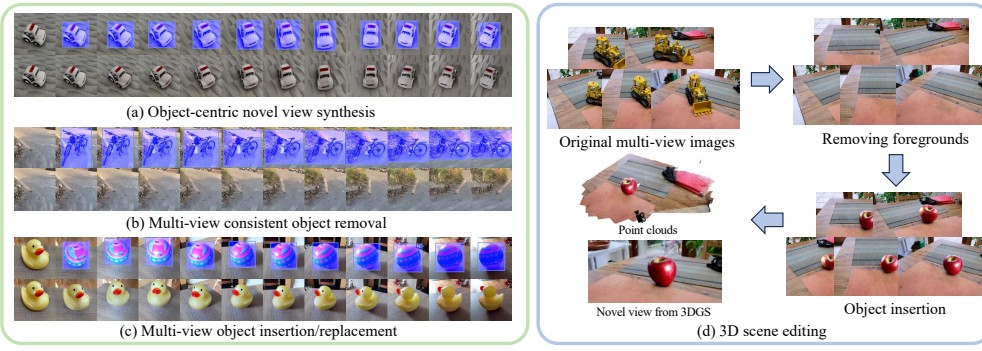

Figure 1: MVInpainter addresses 2D/3D editing tasks: (a) novel view synthesis, (b) multi-view object removal, and (c) object insertion and replacement through multi-view consistent inpainting ability. Given one inpainted or edited reference image, MVInpainter spreads it to other masked views without pose conditions. (d) MVInpainter could be applied to real-world 3D scene editing for dense point clouds by Dust3R [76] or Multi-View Stereo (MVS) [9] and 3DGS [35] with consistent generation.

*difficulty generalizing across various categories.* Even when enhanced by diffusion models, existing NVS-based methods [12, 73, 29, 1] only work in specific scenarios and often fail to generalize to diverse or unseen categories in scene data. 3) *Instance-level 3D editing is time-consuming.* Approaches like instance-level 3D editing [26, 68, 3] and warp-and-inpaint NVS [23, 30, 97] integrate priors from *single-view* T2I models [59], requiring costly dataset updates to address multi-view inconsistency. 4) *Heavy reliance on explicit camera poses.* All methods mentioned rely on accurate camera poses for both training and inference, limiting scalability with pose-free data and broader applicability, particularly in scenarios like short video editing where detailed poses may be unavailable.

To this end, we present a novel perspective to enable 3D editing with a *multi-view consistent inpainting manner.* By starting with an edited 2D reference image from any 2D generative models [98, 14] and applying our method to masked sequences from other viewpoints, we achieve consistent multi-view inpainting results, effectively extending 2D generation into 3D scenarios. The key idea behind this inpainting approach is to focus on seamlessly synthesizing local regions rather than generating entirely new views. Many contextual cues like illumination, shadow, and camera motions are implicitly captured in unmasked regions, which can be activated by foundational T2I models. Thus, our method provides an end-to-end multi-view synthesis solution without requiring test-time optimization or explicit pose conditions, opening up possibilities for both 2D and 3D editing.

Formally, this paper introduces MVInpainter, a multi-view consistent inpainting model built upon a pre-trained StableDiffusion (SD) inpainting model [59]. We incorporate domain adapters and motion modules [25] to MVInpainter as video priors for multi-view consistent structures. Moreover, to encourage appearance consistency, we propose the Reference Key&Value concatenation (Ref-KV), spatially concatenating the key and value features from the reference view to target ones in self-attention modules. Furthermore, MVInpainter operates without explicit poses, utilizing slot-attention mechanisms [45, 87] of encoding and grouping motion priors from unmasked surroundings' optical flow for implicitly pose control. We train two MVInpainters sharing the same pre-trained SD backbone in object-centric (CO3D [57], MVImgNet [95]) and forward-facing (Scannet++ [89], Real10k [103], DL3DV [41]) scenes, respectively. Experiments on unseen scenes and zero-shot datasets as Omni3D [6] and SPInNeRF [51] show the efficacy of MVInpainter in various applications, such as multi-view object removal, synthesis, insertion, and replacement.

In summary, 1) we present MVInpainter as the first multi-view consistent inpainting model to bridge 2D and 3D scene editing. 2) MVInpainter is a pose-free end-to-end approach with high-level flow-based motion control from unmasked regions. 3) MVInpainter largely simplifies the NVS difficulty, which can be generalized to all categories of in-the-wild CO3D, MVImgNet, and Omni3D datasets, as well as complicated forward-facing scenes of Scannet++, Real10k, DL3DV, and SPInNeRF.

## 2 Related Work

**Image Inpainting** devoted to filling masked regions of the image with vivid textures and structures as a long-standing challenge in computer vision, which has been widely investigated in both classical and

learning-based methods [83]. Compared to the low-level feature-based traditional manners [61, 36], learning-based ones gradually dominate this field and achieved substantial successes based on GANs [53, 100, 37, 8], attention mechanism [93, 90], adapted convolutions [42, 94, 70], and diffusion models [47, 62, 59]. Moreover, the image inpainting task can be further extended to reference-guided inpainting and video inpainting. Reference-guided inpainting completes the target image based on one or several reference images, which incorporates 3D information for accurate structures [105, 101] or T2I priors for examplar-based recovery and editing [86, 14, 7]. On the other hand, video inpainting methods [24, 39, 102] often include the optical flow to capture motions for superior spatial and temporal coherence. However, we should clarify that the aforementioned inpainting methods cannot achieve multi-view consistent inpainting for 3D scene editing. The reference-based manners lack multi-view consistency across all generated views, while video-based methods focus on moving foregrounds rather than the synthesis with large viewpoint changes as verified in our experiments.

**3D Generation.** Given text descriptions or reference images, 3D generative models produce high-quality 3D assets, which have been widely investigated recently, benefiting from the rapid development of diffusion-based 2D T2I models [59, 63, 4, 54, 21]. Some pioneering works with score distillation sampling (SDS) loss [75, 55] leverage priors from diffusion-based 2D supervision for the 3D generation, which is further explored to better optimization objectives [106, 79] and multi-stage learning [40, 13, 71]. On the other hand, Zero123 [44] fine-tuned the T2I model for object-level NVS, which is further investigated for consistent multi-view synthesis [67, 66, 43, 46]. Besides, enhanced by the good 3D feature presentations, like tri-plane [11], and scalable network backbones [74], foundational 3D generation models also achieved impressive results [109, 85, 31], training with extremely large 3D datasets from scratch. However, all these works are trained with large-scale synthetic 3D objects or optimized through SDS without any interaction with complicated backgrounds, making it difficult to generalize to real-world editing scenarios with reasonable illustrations and shadows.

**Novel View Synthesis (NVS).** Before the diffusion models, most NVS works focused on learning promising feature encoder [69, 91] with blur regression-based predictions. After the development of 3D-aware diffusion models [12, 73, 1] and fine-tuning from foundational T2I models [64, 29, 82], NVS results are significantly improved. However, generating whole novel views is too challenging. Thus these methods still suffer from constrained generalization with seen categories or expensive test-time optimization. Another way is to iteratively warp and inpaint the novel views through monocular depth estimation and 2D inpainting [92, 23, 30, 97]. Despite impressive scene-level synthesis results, these methods suffer from prohibitive time costs for the warp-and-inpaint dataset update. Moreover, ambiguous depth estimations would degrade the structures of foreground objects.

**NeRF and 3DGS Editing.** With the development of NeRF [49] and 3DGS [35], many follow-ups tried to integrate them into 2D generative models, including NeRF inpainting [51, 81, 50], textual-guided semantic editing [26, 15] and local editing [108, 3, 38] based on SDS loss. InseRF [65] unifies both NeRF editing and 3D generation by inserting an image-to-3D generation into multi-view images for the NeRF optimization. Unfortunately, local editing-based approaches suffer from unnatural and disharmonious results, especially for the illumination and shadow in boundary regions. Furthermore, all these works require scenes with accurate camera poses and costly test-time optimization to encourage multi-view consistency, limiting their real-world applications.

## 3  Approach

**Overview.** We show the overall pipeline and contributions in Fig. 2, which is detailed in Fig. 3(a). The inputs of MVInpainter are sequential images $\mathbf{I}^{0:N}$ of the same scene and related masks $\mathbf{M}^{0:N}$ with $N+1$ total views; $\mathbf{I}^0$ indicates the clean reference image manipulated by any 2D editing approach, while $\mathbf{I}^{1:N}$ are other target views needed to be inpainted by MVInpainter with consistent results.

**Multi-View Inpainting Formulation.** We focus on multi-view inpainting rather than naive NVS as it suits real-world editing better, because 1) most real-world edits do not need to synthesize complete novel views; 2) the inpainting formulation activates the inherent in-context priors from T2I models to ensure natural illumination and shadow, which are intractable to be addressed by 3D generation trained with synthetic data; 3) such a simplified formulation alleviates the training difficulty and the dependency on camera poses. Specifically, MVInpainter enjoys consistent multi-view inpainting to bridge 2D and 3D editing, which is built upon a foundation 2D inpainting model, SD1.5-inpainting [59] with two data prerequisites. First, the input multi-view images $\mathbf{I}^{0:N}$ should

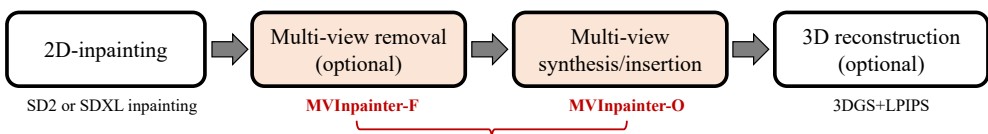

Figure 2: The overall pipeline and main contributions of MVInpainter. We primarily focus on multi-view inpainting, while the 3D reconstruction is detailed in Appendix Sec. C.

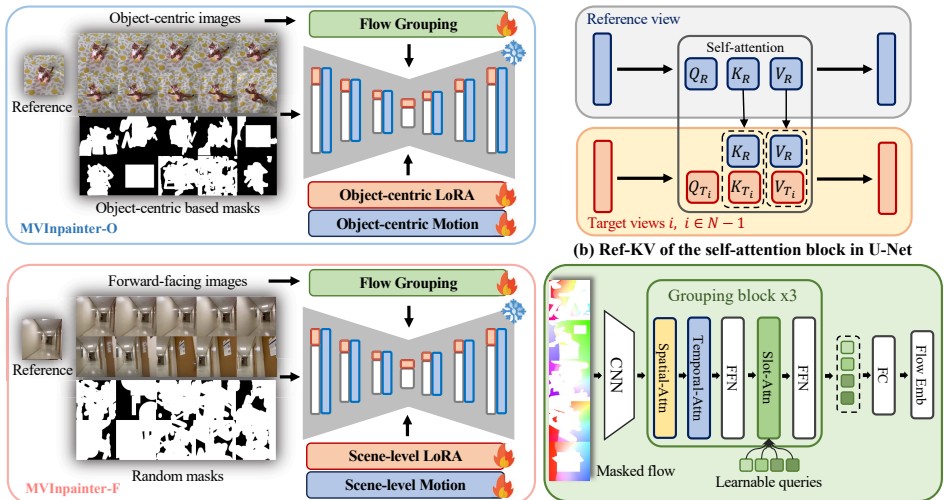

Figure 3: (a) The overview of the proposed MVInpainter. MVInpainter-O is trained on object-centric data, while MVInpainter-F is trained on forward-facing data with a shared SD-inpainting backbone of different LoRA/motion weights and masking strategies. The object-centric MVInpainter focuses on the object-level NVS, while the forward-facing one is devoted to object removal and scene-level inpainting. (b) The Ref-KV is used in spatial self-attention blocks of denoising U-Net. (c) The slot-attention based flow grouping module is used to learn implicit pose features. Dashed boxes in (b) and (c) mean feature concatenation.

be captured in an ordered camera trajectory to alleviate the pose requirement. Second, $\mathbf{M}^0$ is a zero matrix to ensure the first view is clean without masking, while other masks $\mathbf{M}^{1:N}$ should cover all possible regions in respective views where the target object would be placed at. To elegantly meet this demand, we propose a heuristic masking technique for the inference detailed in Sec. 3.4. Hence, the input of MVInpainter could be formulated as an ordered video sequence with $(N + 1)$ frames as:

$$x_t = [z_t^{0:N}; z(1 - \hat{\mathbf{M}})^{0:N}; \hat{\mathbf{M}}^{0:N}] \in \mathbb{R}^{(N+1) \times H \times W \times 9}, \tag{1}$$

where $t$ indicates the timestep in the diffusion; $z_t^{0:N}$ denote the 4-channel noised latent feature of $\mathbf{I}^{0:N}$ after the VAE encoding [59]; $\hat{\mathbf{M}}^{0:N}$ and $z(1 - \hat{\mathbf{M}})^{0:N}$ mean that the downsampled masks and noise-free latent features in unmasked regions are always concatenated as the input condition. We learn MVInpainter through the epsilon prediction $\epsilon_\theta$ [28], while the MSE loss can be written as:

$$\mathcal{L} = \mathbb{E}_{x_0, \epsilon \sim \mathcal{N}(0,1), c, t} \left[ \|\epsilon - \epsilon_\theta(x_t, \tau_\phi(c), t)\|^2 \right], \tag{2}$$

where $c$ denotes the prompt text encoded by the textual CLIP $\tau_\phi$ [56].

## 3.1 MVInpainter Tasks

Here we tackle two related but distinct tasks: multi-view consistent object removal, and object insertion or replacement. Due to the differing challenges of each task, we employ the same structure, called MVInpainter, but trained with different multi-view data, prompts, and masking strategies, leading to two variants: MVInpainter-O and MVInpainter-F.

**Object-Centric and Forward-Facing Datasets.** Particularly, we train these two MVInpainters to handle different distributions of multi-view data: *object-centric* and *forward-facing* data. Object-

centric datasets (CO3D [57], MVImgNet [95], Omni3D [6])[2] feature a single object at the center of all views, captured with the camera circling around it. On the other hand, forward-facing datasets (Real10K [103], Scannet++ [89], DL3DV [41]) resemble static video data with camera movements but no specific foreground objects. Essentially, object-centric images heavily favor a single foreground object in all views, so an MVInpainter trained on them struggles to reliably remove objects without hallucinating any artifacts. Conversely, forward-facing images are not conducive to modeling multi-view object synthesis with significant viewpoint changes.

To address these challenges, we separately train MVInpainter-O and MVInpainter-F on object-centric and forward-facing datasets, respectively. They involve different LoRAs, motion modules (Sec. 3.2), and flow grouping modules (Sec. 3.3). Note that we keep the SD-inpainting backbone frozen for both models. While fine-tuning the entire SD backbone for MVInpainter-O might help convergence with mixed CO3D and MVImgNet datasets, we maintain consistent settings with MVInpainter-F for simplified discussion in this study. Please refer to Sec. B.5 of Appendix for more details.

**Prompt.** For MVInpainter-O, we empirically find that meaningful prompts $c$ benefit to preserve the identity and appearance of objects, as these prompts provide complementary information for the object synthesis with unseen viewpoints. Thus we utilize InternLM [19] to extract captions for training and inference. For MVInpainter-F which is mainly used for object removal and scene completion with minor viewpoint changing, we leverage the prompt tuning technique [7] with 16 trainable tokens as the global prompts instead of specific descriptions for stable generation.

**Masking Strategy.** We adopt hybrid inpainting masks [70, 8] for MVInpainter, including random irregular, rectangular, and segmentation-based masks. In particular, we further employ the object-level tracking masks for MVInpainter-O training, which could be easily accomplished by SAM-tracking [88]. Moreover, to avoid the mask overfitting towards the object shape, we follow [7] to randomly disturb the object masks with rectangular and irregular masks. Note that MVInpainter-O preserves a little percentage (15%) to use random masks without the object ones to encourage the cross-view learning ability for the frame interpolation and sparse-view NVS (Appendix Sec. B.2).

## 3.2 Multi-View Consistent Inpainting Model

**Motion Priors from Video Models.** Since the input of MVInpainter could be regarded as the video sequence as in Eq. 1, it is intuitive to leverage motion priors from video models to improve performance. Thus we employ the domain adapted LoRA [32] and temporal transformers from AnimateDiff [25] pre-trained on video data as the initialized motion parts of MVInpainter. Although AnimateDiff is not trained for an inpainting model, we surprisingly find that it could be well converged with only a few hundred steps of fine-tuning, while both MVInpainter and AnimateDiff share the same SD1.5 backbone. Empowered by motion priors, MVInpainter achieves significantly superior structural consistency as discussed in Sec. 4.4.

**Reference Key&Value Concatenation (Ref-KV).** Inspired by [66, 33], to further ensure appearance consistency, we adopted Ref-KV in the self-attention of denoising U-Net to activate the inherent capacity of T2I models. Ref-KV spatially concatenates reference features to target keys and values to inject appearance guidance during attention aggregation as in Fig. 3(b). We clarify the originality of Ref-KV as follows. Compared to aggregating all frames [29], Ref-KV only focuses on the first reference view, which reduces memory and computational costs, and also substantially enhances the appearance consistency as verified in Fig. 7. Different from the reference attention in [66, 33], MVInpainter is an inpainting model, which always captures the unmasked reference latent without noise (Eq. 1), thus it is unnecessary to re-scale the latent by multiplying a large scale [66] or use the noise-free latent from another U-Net [33].

## 3.3 Pose-Free Flow Grouping

Benefiting from the inpainting formulation and the ordered input sequence of MVInpainter, our approach could be trained and tested without explicit camera poses. However, it is still non-trivial to inpaint the foreground object with correct poses while the masks are large or the unmasked environment is ambiguous and textureless. To overcome this, we leverage the low-level optical flows

---

[2]This paper only focuses on data captured by object-centric camera trajectory rather than discovering decoupled observations through unsupervised feature learning [45, 104, 34].

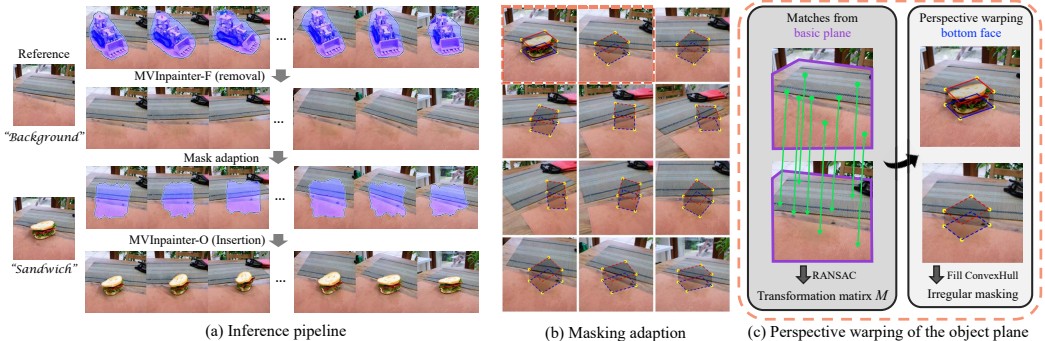

| (a) Inference pipeline | (b) Masking adaption | (c) Perspective warping of the object plane |

Figure 4: (a) The inference pipeline includes object removal, mask adaption, and object insertion. (b) The illustration of heuristic masking adaption, which is built from yellow points of the closed convex hull. (c) The perspective warping based on the basic plane and the bottom face. All matches are on the basic plane filtered by Grounded-SAM [58] with captions "table" and "tablecloth".

extracted by RAFT [72] to guide the MVInpainter generation. We regard the reverse $N$-frame optical flow as $\mathbf{F}^{0:N-1} \in \mathbb{R}^{(N-1) \times H \times W \times 2}$. All flow inputs are masked to avoid leakage[3].

In the pilot study, we first applied the 2D CNN and self-attention modules to encode the flow features and added them to the U-Net input as additional conditions. But such simple incorporation failed to improve the MVInpainter as the 'dense flow' setting in Tab. 3b. Because such an explicit dense flow injection leads to the overfitting pitfall, *i.e.*, MVInpainter is strongly controlled by the flow inputs and ignores other contextual clues learned from the foundational T2I model. We hope the flow feature should carry more high-level motion characters, such as the direction and speed of the camera trajectory rather than the detailed correlation. Thus we propose the flow grouping enhanced by the slot-attention [45, 87] as shown in Fig. 3(c).

Formally, slot-attention maintains $K$ learnable query vectors as $\mathbf{Q} \in \mathbb{R}^{K \times d}$, where $d$ denotes the channels; $K = 4$ in this paper. Key and value are the same flow features as $\mathbf{K} = \mathbf{V} \in \mathbb{R}^{HW \times d}$. Then the slot-attention can be formulated as:

$$\text{Slot-Attn}(\mathbf{Q}, \mathbf{K}, \mathbf{V}) = softmax_q(\frac{\tilde{\mathbf{Q}}\tilde{\mathbf{K}}^T}{\sqrt{d}})\tilde{\mathbf{V}} \in \mathbb{R}^{K \times d}, \tag{3}$$

where $\tilde{\mathbf{Q}}, \tilde{\mathbf{K}}, \tilde{\mathbf{V}}$ indicate $\mathbf{Q}W_q, \mathbf{K}W_k, \mathbf{V}W_v$ encoded by linear weights $W_q, W_k, W_v \in \mathbb{R}^{d \times d}$. Note that the slot-attention is similar to the cross-attention except that the former should be normalized in the query dimension, while the latter is normalized in the key dimension. As the global queries, all slot features $\mathbf{Q}$ enjoy the high-level motion information aggregated by dense flow features. We take these slot features and subsequently use an FC layer to encode them into a single flow grouping embedding as shown in Fig. 3(c). According to the ablation study in Tab. 3b, we concatenate this embedding to the CLIP feature as an extra motion 'token' with comparable performance and less trainable weights. Moreover, we found that using temporal 3D attention for the flow grouping can further improve the results, while the slot features are shared across all views with more general information. Flow grouping achieves more robust guidance for MVInpainter than the dense flow features, even with some corrupted or inaccurate flow inputs.

## 3.4 Inference

We show the inference pipeline of MVInpainter in Fig. 4(a). Specifically, given sequential images, MVInpainter could address the general multi-view editing through object removal and object insertion mentioned in Sec. 3.1. The object removal stage can be omitted if there are no foreground objects (inserting only) or the target object shares similar masks as the original one (replacing). For the object removal, we utilize the MVInpainter-F based on the reference image inpainted by SD-inpainting with the caption "background". For the object insertion, the reference image can be achieved by any 2D

---

[3]We extract flows before masking, because foregrounds largely benefit the flow quality. We further dilate masks with 5 pixels for flow to avoid leakage. As low-level local features, no conflict is observed when using these masked flows for applications like removal and replacement.

generative models, such as T2I inpainting [59, 77, 84] and exemplar-based inpainting [86, 14]. Then MVInpainter-O is leveraged to expand the single-view generation to the multi-view scenario. We propose a heuristic masking adaption to tackle the most critical issue, *i.e.*, building suitable masks without any pose conditions before the insertion.

**Masking Adaption** is based on points from the closed convex hull as shown in Fig. 4(b), which can be obtained from the 3D bounding box of the foreground object. We could use open-vocabulary 3D grounding [16] or manually annotate four landmarks to achieve the bottom face and estimate the top face through the height of the 2D mask. To reasonably warp the 3D box, our masking adaption should meet an important and easily satisfied condition: *the 3D box's bottom face and the basic plane on which the object is placed must be the same plane.* Intuitively, the bottom face should be close to the ground. Therefore, the basic plane and the bottom face of the 3D box from different image pairs share the same perspective transformation matrix. Thanks to the dense matching [20], it is easy to obtain dense matching pairs on the basic plane through the Grounded-SAM [58] as in Fig. 4(c). Subsequently, we achieve the transformation matrix $M$ by RANSAC [22] with 100 sampled matching pairs and apply the perspective warping for the new bottom face. Note that this perspective warping cannot be used on the top face, because the top face is just parallel to the basic plane rather than close to it. Instead, we get the new top face following the bottom landmarks with a constant height from the 2D mask. Finally, we fill the convex hull and optionally mask it with irregular brushes as in [7] or dilate the 3D box masks. The masking adaption is a flexible strategy, which not only locates the position of the object across different views but also incorporates human priors by manually annotating for some objects with very special shapes, such as the baseball bat in Appendix Fig. 13.

## 4 Experiments

**Datasets.** MVInpainter-O is trained on the object-centric data that includes full categories of CO3D [57] and MVImgNet [95]. Moreover, we regard the Omni3D [6] as the zero-shot validation. MVInpainter-F is trained on the forward-facing data with Real10K [103], Scannet++ [89], and DL3DV [41], including both indoor and outdoor scenes. We further employ comparison on SPInNeRF [51] to verify the object removal ability. To mitigate the imbalanced category distribution, we sample an equivalent subset of scenes for each category in every epoch. All images are resized and cropped into 256×256 for both inpainting and flow extraction. More details about the dataset are discussed in Appendix Sec. B.1. To the best of our knowledge, MVInpainter is the first scene-level generative model that could be generalized on all categories of both CO3D and MVImgNet.

**Dynamic Frame Sampling.** To alleviate the training costs from long sequences, we first train MVInpainter with $(N + 1) = 12$ frames. Then, only a few steps (1/10) of fine-tuning with dynamic frame numbers from $(N + 1) \in [8, 24]$ are sufficient for a good frame number adaption. For the inpainting of more frames, frame interpolation-based inpainting introduced in Appendix Sec. B.2 should be considered. Besides, we also randomly sample the frame interval to encourage generalization.

**Training Setup.** All trainings are accomplished on 8 A800 GPUs. We train MVInpainter-O and MVInpainter-F for 100k and 60k steps with batch size 64, frame number 12, learning rate 1e-4 for 3 days and 2 days respectively. Then we fine-tune the model with dynamic frames for 10k steps.

**Metrics.** In this paper, we evaluate our method with PSNR, LPIPS [99], FID [27], and KID [5]. We also include CLIP score [56] for the NVS task to verify the identity maintenance capability. For the object removal, we further compare the similarity of DINOv2 features [52] extracted from masked regions to evaluate the inpainting consistency, denoting average DINOv2 similarity (DINO-A) and minimal DINOv2 similarity (DINO-M) among masked patches, respectively.

### 4.1 Object-Centric Results

Results of object-centric data are compared in Tab. 1 and Fig. 5. We considered two types of approaches, including NVS manners: ZeroNVS [64]; inpainting-based methods: Nerfiller [80] and LeftRefill [7]. Note that ZeroNVS requires explicit camera poses. Besides, the inpainting-based methods are pose-free with enlarged bounding box masks to avoid masking shape leakage. Without the inpainting formulation, the scene-level ZeroNVS struggles to directly synthesize novel views without SDS. For inpainting-based approaches, LeftRefill can capture the scene's main object but fails to maintain multi-view consistency. Nerfiller only retains the shape of several views, while

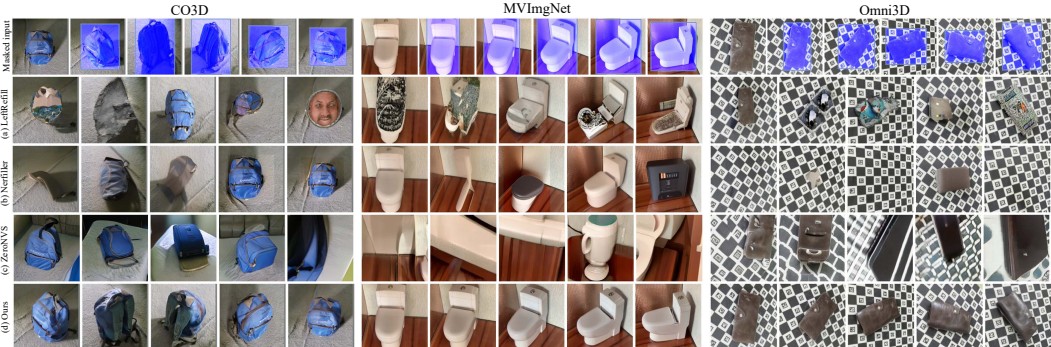

Figure 5: Object-centric results on CO3D, MVImgNet, and Omni3D. The first row denotes the reference (first column) and other masked inputs, while other results are sampled from LeftRefill [7], Nerfiller [80], ZeroNVS [64], and our MVInpainter. Please zoom-in for details.

Table 1: Quantitative results of object-centric NVS with enlarged bounding box masks compared on CO3D [57] and MVImgNet [95]. We also include Omni3D [6] as a zero-shot test set without being trained by any competitor. All KID results are multiplied by 100.

|  | CO3D+MVImgNet | | | | | Omni3D (zero-shot) | | | | |
|---|---|---|---|---|---|---|---|---|---|---|
| ZeroNVS [64] | 12.44 | 0.606 | 41.90 | 0.981 | 0.6028 | 9.38 | 0.627 | 82.81 | 5.421 | 0.5451 |
| Nerfiller [80] | 18.29 | 0.310 | 36.64 | 0.491 | 0.6603 | 16.10 | 0.272 | 37.04 | 1.056 | 0.6279 |
| LeftRefill [7] | 17.74 | 0.283 | 38.06 | 0.826 | 0.6392 | 17.09 | 0.239 | 27.81 | 0.775 | 0.6484 |
| Ours | **20.25** | **0.185** | **17.56** | **0.154** | **0.8182** | **19.19** | **0.153** | **16.40** | **0.345** | **0.7667** |

other views suffer from significantly inferior structures and identities. Our model outperforms all competitors with prominent achievements on both the in-domain test set and the zero-shot Omni3D.

## 4.2 Forward-Facing Results

**Object Removal.** We quantitatively verify the object removal ability of MVInpainter-F on SPInNeRF test set [51] in the left of Tab. 2, while qualitative comparisons of the train set are shown in Fig. 8 of Appendix. Other competitors include conventional single-view based inpainting manners: LaMa [70], MAT [37], and SD-inpainting [59]. We further compare the pose-free reference-inpainting Left-Refill [7] and the video inpainting manner ProPainter [102]. Formally, MAT and SD-inpainting suffer from inconsistent inpainting results with poor DINO-A and DINO-M. Though LaMa achieves the highest DINO-M, it generates consistent blur and artifacts as in Appendix Fig. 8(a). For the reference-based manners, LeftRefill only conditions on the first view without multi-view consistency, while ProPainter performs inferior in synthesis quality. Since the small test set, FID would be largely degraded if one result contains artifacts. Our method enjoys the best performance in LPIPS, FID, and DINO-A with consistent generations without any unstable hallucination.

**Scene-Level Inpainting.** We provide quantitative results of multi-view scene inpainting in the right of Tab. 2 corrupted with large irregular masks. Our method achieves the best results of all metrics, prominently outperforming video-based inpainting, reference-guided inpainting, and other single-view inpainting approaches.

## 4.3 Real-World 3D Scene Editing

We verify the in-the-wild scene editing ability of MVInpainter following Sec. 3.4 in Fig. 6, where the background images are from the unseen MipNeRF360 [2]. MVInpainter-F achieves stable and consistent object removal, and MVInpainter-O performs high-quality multi-view generation for various object shapes based on the flexible mask adaption. Benefiting from the consistent results, our method enjoys reliable reconstruction with stereo-based Dust3R [76] and MVS [9]. Sec. C of the Appendix discusses more details about the point cloud and 3DGS reconstruction.

Table 2: Quantitative results of scene-level forward-facing NVS with masks. The clean SPIn-NeRF [51] dataset with consistent object masks is used to evaluate the object removal, while the unseen scenes from Scannet++ [89], Real10k [103], and DL3DV [41] degraded by random masks are used to verify the basic inpainting ability. All KID results are multiplied by 100.

| | SPInNeRF (removal) | | | | | Scannet+Real10K+DL3DV (inpainting) | | | |
| --- | --- | --- | --- | --- | --- | --- | --- | --- | --- |
| | PSNR↑ | LPIPS↓ | FID↓ | DINO-A↑ | DINO-M↑ | PSNR↑ | LPIPS↓ | FID↓ | KID↓ |
| LaMa [70] | 28.62 | 0.054 | 15.26 | 0.8909 | **0.6019** | 17.61 | 0.337 | 38.47 | 0.981 |
| MAT [37] | 27.05 | 0.067 | 28.81 | 0.8727 | 0.5760 | 15.47 | 0.377 | 37.38 | 0.899 |
| SD-inpaint [59] | 26.98 | 0.070 | 19.32 | 0.8556 | 0.4422 | 13.54 | 0.417 | 38.67 | 1.048 |
| LeftRefill [7] | 30.29 | 0.102 | 18.02 | 0.8931 | 0.5652 | 15.14 | 0.380 | 38.06 | 1.334 |
| ProPainter [102] | **31.72** | 0.047 | 12.25 | 0.8757 | 0.5534 | 20.42 | 0.306 | 61.76 | 2.642 |
| Ours | 28.87 | **0.036** | **7.66** | **0.8972** | 0.5937 | **20.91** | **0.173** | **15.58** | **0.252** |

Table 3: Ablation studies on CO3D. 'w.o. inp' means the baseline without the inpainting formulation.

| | PSNR↑ | LPIPS↓ | CLIP↑ |
| --- | --- | --- | --- |
| Baseline | 17.16 | 0.305 | 0.750 |
| Baseline (w.o. inp) | 14.35 | 0.443 | 0.648 |
| +AnimateDiff | 17.31 | 0.308 | 0.756 |
| +Ref-KV | 17.90 | 0.283 | 0.773 |
| +Object mask | 18.64 | 0.250 | 0.796 |
| +Flow emb | **18.93** | **0.240** | **0.798** |

(a) Ablation results of different proposed components

| | PSNR↑ | LPIPS↓ | CLIP↑ |
| --- | --- | --- | --- |
| No Flow | 18.64 | 0.250 | 0.796 |
| Dense Flow | 18.53 | 0.247 | 0.792 |
| Slot2D Flow (time-emb) | 18.74 | 0.244 | **0.798** |
| Slot2D Flow (cross-attn) | 18.81 | 0.245 | 0.796 |
| Slot3D Flow (cross-attn) | **18.93** | **0.240** | **0.798** |

(b) Ablation of various strategies to inject flow guidance

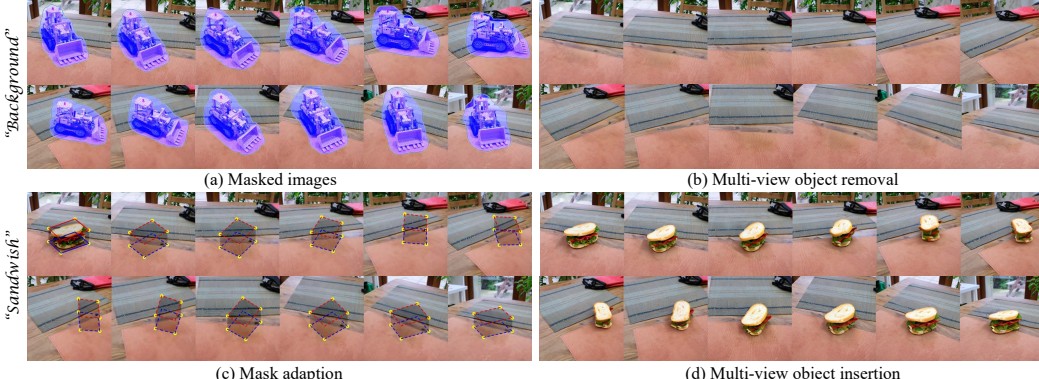

*"Background"*

(a) Masked images      (b) Multi-view object removal

*"Sandwish"*

(c) Mask adaption      (d) Multi-view object insertion

Figure 6: Results of multi-view scene editing, including (b) multi-view object removal, (c) mask adaption, (d) multi-view object insertion. More results are shown in Fig. 13 and Fig. 14 of Appendix.

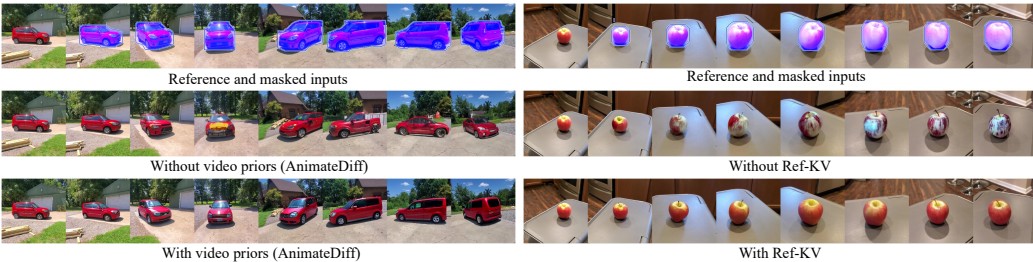

Reference and masked inputs      Reference and masked inputs

Without video priors (AnimateDiff)      Without Ref-KV

With video priors (AnimateDiff)      With Ref-KV

Figure 7: Qualitative ablation studies of AnimateDiff initialization and Ref-KV on CO3D.

### 4.4 Ablation Study

To verify the effectiveness of each component from MVInpainter, we conduct ablation studies on CO3D as in Tab. 3. From Tab. 3a, the baseline is built upon SD1.5-inpainting with random-initialized LoRA and motion transformer blocks, and we also compare the alternative baseline without the inpainting formulation built upon naive SD1.5. The multi-view inpainting formulation largely improves the results, and more details about the inpainting comparison are discussed in Appendix Sec. B.4. Besides, we show qualitative ablation results in Fig. 7, which indicate that video priors from AnimateDiff and Ref-KV could substantially facilitate the structure and appearance consistency respectively. We also realize that object-level tracking masks are critical for training MVInpainter-O. Moreover, we analyze the effect of flow grouping in Tab. 3b. Without any flow guidance, our method fails in some ambiguous cases, such as large pose changes of stop signs and laptops in Fig. 12 of Appendix, while dense flow slightly hinders the identity (PSNR and CLIP). The proposed slot-attention based flow grouping outperforms the vanilla flow injection, while incorporating flow embedding by cross-attention is more lightweight with comparable performance. Further, the flow grouping can be improved with 3D temporal attention learning.

## 5 Conclusion

This paper proposes MVInpainter, a multi-view consistent inpainting method to expand 2D generations into 3D scenes by multi-view object removal, insertion, and replacement. MVInpainter enjoys a pose-free inpainting formulation built upon the SD-inpainting backbone with motion modules. Motion initialization based on video priors and Ref-KV are presented to facilitate the structure and appearance consistency respectively. Furthermore, we propose to use flow grouping based on the slot-attention to encourage implicit motion control. For the inference, we present a novel mask adaption strategy to warp object masks to novel views. Sufficient experiments on both object-centric and forward-facing datasets verified the effectiveness of MVInpainter.

**Acknowledgements.** We would like to thank Yanwei Fu. Dr. Fu is with School of Data Science in Fudan, Fudan ISTBI—ZJNU Algorithm Centre for Brain-inspired Intelligence, Shanghai Key Lab of Intelligent Information Processing, and Technology Innovation Center of Calligraphy and Painting Digital Generation, Ministry of Culture and Tourism, China. The computations in this research were supported by the CFFF platform of Fudan University.

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

# A Supplementary Results

We provide more qualitative results in this section, including object removal results of Fig. 8 and Fig. 9; multi-view scene editing of Fig. 13 and Fig. 14; the generalization of the proposed mask adaption of Fig. 15; results of multi-view object-level NVS in Fig. 10; object replacement by T2I inpainting model and the exemplar-based AnyDoor [14] in Fig. 11. We further provide some qualitative ablation studies in Fig. 12. These visualizations verify the wide application of the proposed MVInpainter.

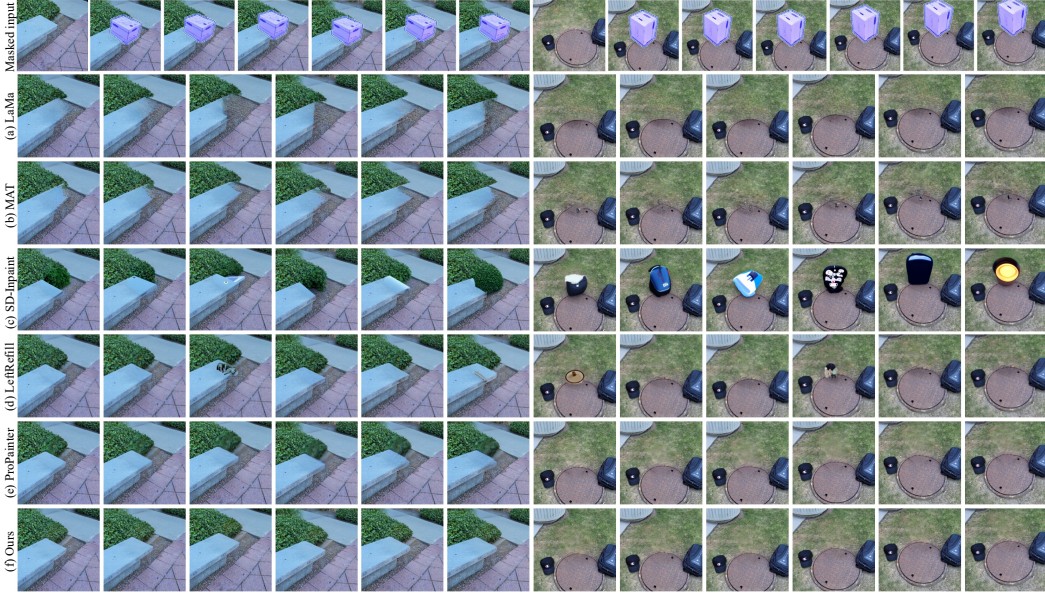

Figure 8: Object removal visualization on SPInNeRF. The first row denotes the reference (first column) and other masked inputs, while other results are sampled from the inpainted sequence.

# B More Details and Comparison

## B.1 Dataset Details

We list detailed data information in Tab. 4. For the object-centric training data, CO3D [57] contains 24k sequences with 51 categories, and MVImgNet [95] contains about 210k sequences with 223 categories, while 15 categories of MVImgNet are seen as the zero-shot test set. Note that we discard some images with too large foreground masks. For the forward-facing data, we choose 190 unseen scenes from Real10K [103], Scannet++ [89], and DL3DV [41] as the mixed scene-level validation. To quantitatively verify the object removal, 10 scenes are selected from SPInNeRF [51] test set with consistent object masks and ground truth images without foregrounds. For the quantitative comparison, we sample several test views for each scene at the maximum interval with officially provided consistent object masks. Besides, we also provide qualitative results of these scenes sampled from the training set with objects. All quantitative validations are based on 24 views, while qualitative results are validated with 12 and 24 views separately. To simplify the visualization, our paper just includes representative views rather than showing all views.

Table 4: Details about the training and testing datasets of MVInpainter.

| Datasets | Train | | Test | |
|---|---|---|---|---|
| | Categories | Sequences | Categories | Sequences |
| CO3D [57] | 51 | 24k | 51 | 102 |
| MVImgNet [95] | 223 | 210k | 238 | 238 |
| Omni3D [6] | – | – | 61 | 244 |

(a) Object-centric data

| Datasets | Train Sequences | Test Sequences |
|---|---|---|
| Real10K [103] | 13k | 50 |
| DL3DV [41] | 5900 | 100 |
| Scannet++ [89] | 340 | 40 |
| SPInNeRF [51] | – | 10 |

(b) Forward-facing data

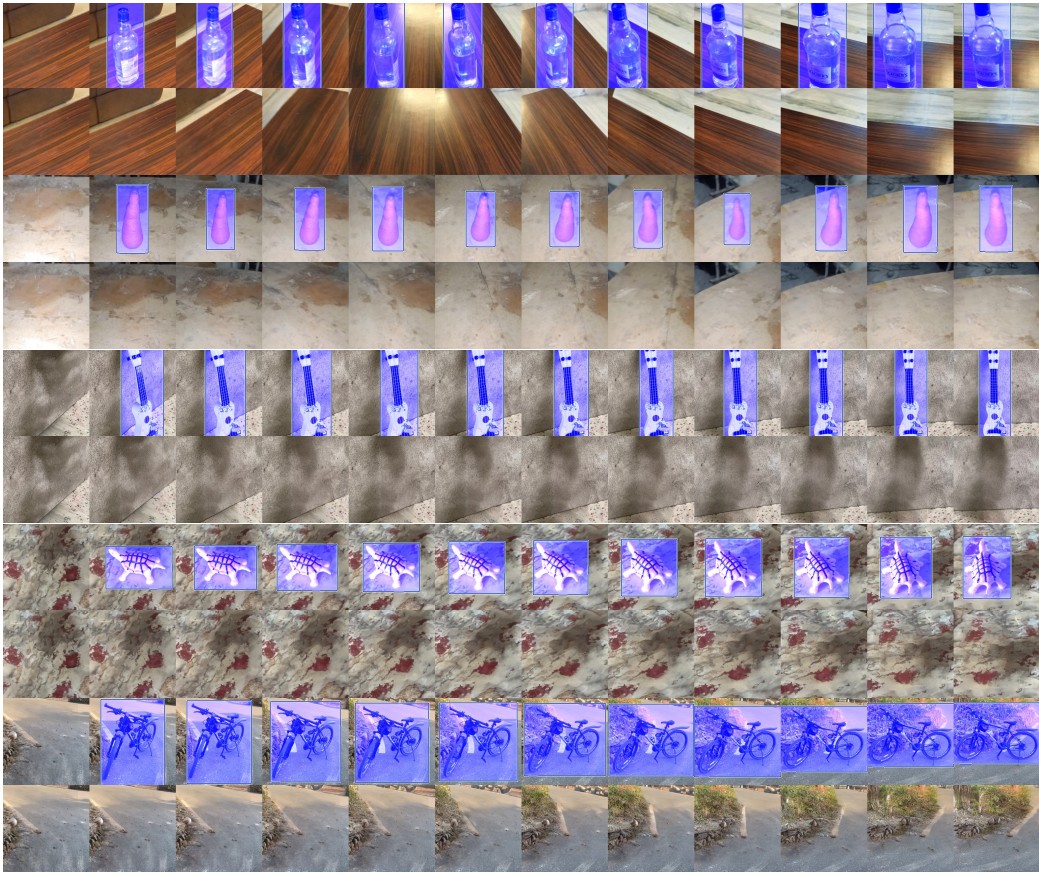

Figure 9: Object removal results from MVInpainter-F on CO3D (rows 1, 2) and MVImgNet (rows 3, 4, 5). The first view of each sequence is inpainted by SD-inpainting with a "background" caption.

## B.2 Frame Interpolation

To inpaint extremely long sequences, MVInpainter could first inpaint some keyframes, then apply the frame interpolation-based inpainting to extend these keyframes to more views. Formally, we first interactively mask and inpaint among keyframes like Fig. 16(a). Afterward, we uniformly sample long-range inpainted results as fixed conditions at left (max to 12), and then interactively inpaint other views to preserve the identity and appearance as Fig. 16(b).

## B.3 Asymmetric VAE Decoder

We found some color difference near the masking boundary in some inpainting results as in Fig. 17, especially for indoor cases with textureless regions. Moreover, we further found that this issue also occurred with the original SD-inpainting. Hence, we follow the asymmetric VAE decoder [107] which includes unmasked image pixels and masks as additional inputs, and apply the data augmentation in [78] to fine-tune the VAE decoder. Though the quantitative results are almost unchanged, the augmented new decoder enjoys much more consistent visualizations as shown in Fig. 17.

## B.4 Detailed Ablation for Inpainting Formulation

To verify the effect of the inpainting formulation, we further detail the comparison in Tab. 5 and Fig. 18. The 'Baseline' is the same as the one listed in Tab. 3a, denoting the backbone of SD1.5-inpainting[4]. Both 'SD-blend' and 'SD-NVS' utilize vanilla SD1.5[5] as the backbone without any

---

[4]`https://huggingface.co/runwayml/stable-diffusion-inpainting`
[5]`https://huggingface.co/runwayml/stable-diffusion-v1-5`

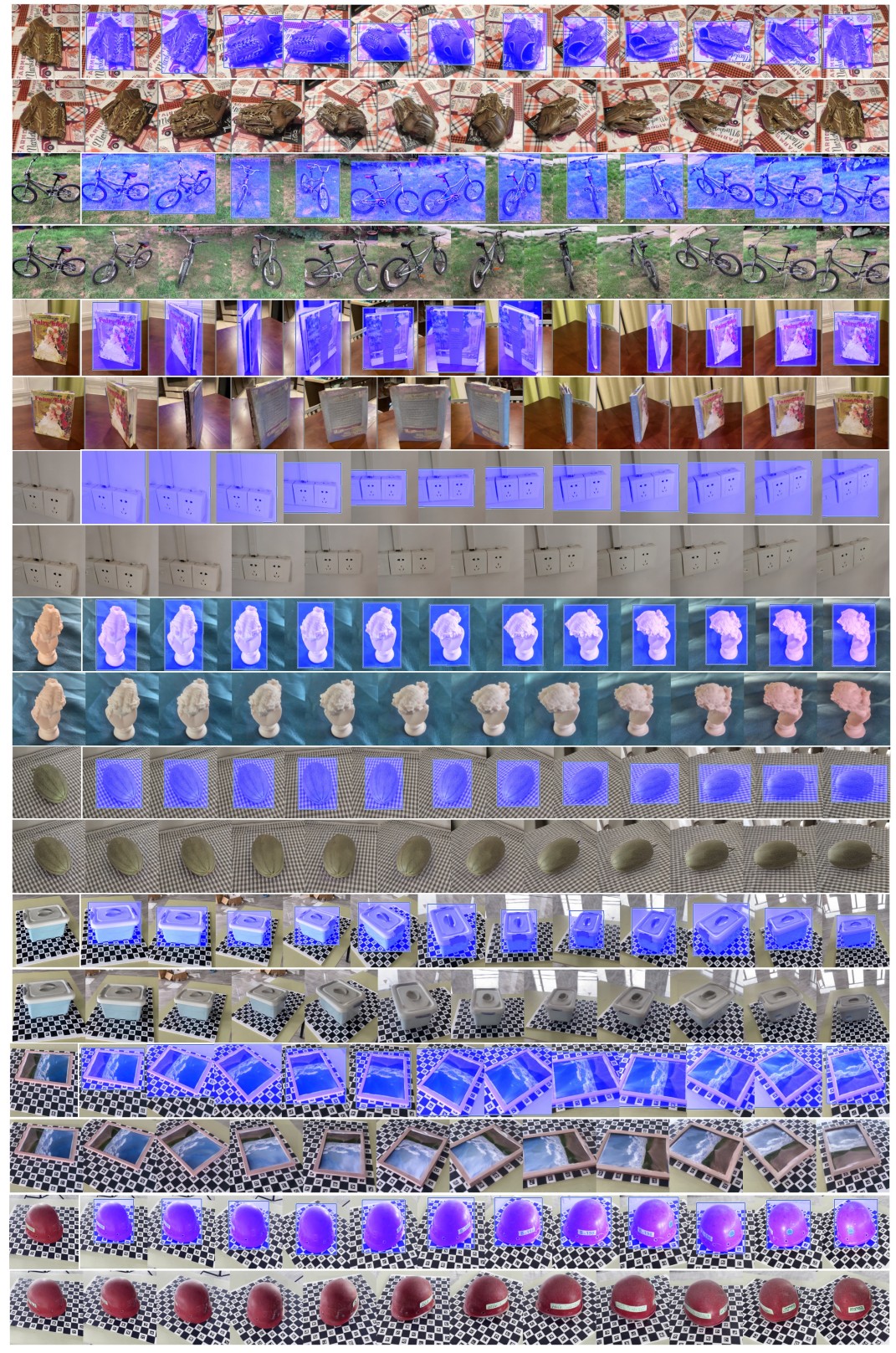

Figure 10: Object-level NVS on CO3D (groups 1 to 3), MVImgNet (groups 4 to 6), and Omni3D (groups 7 to 9). The first row of each group contains an additional reference view.

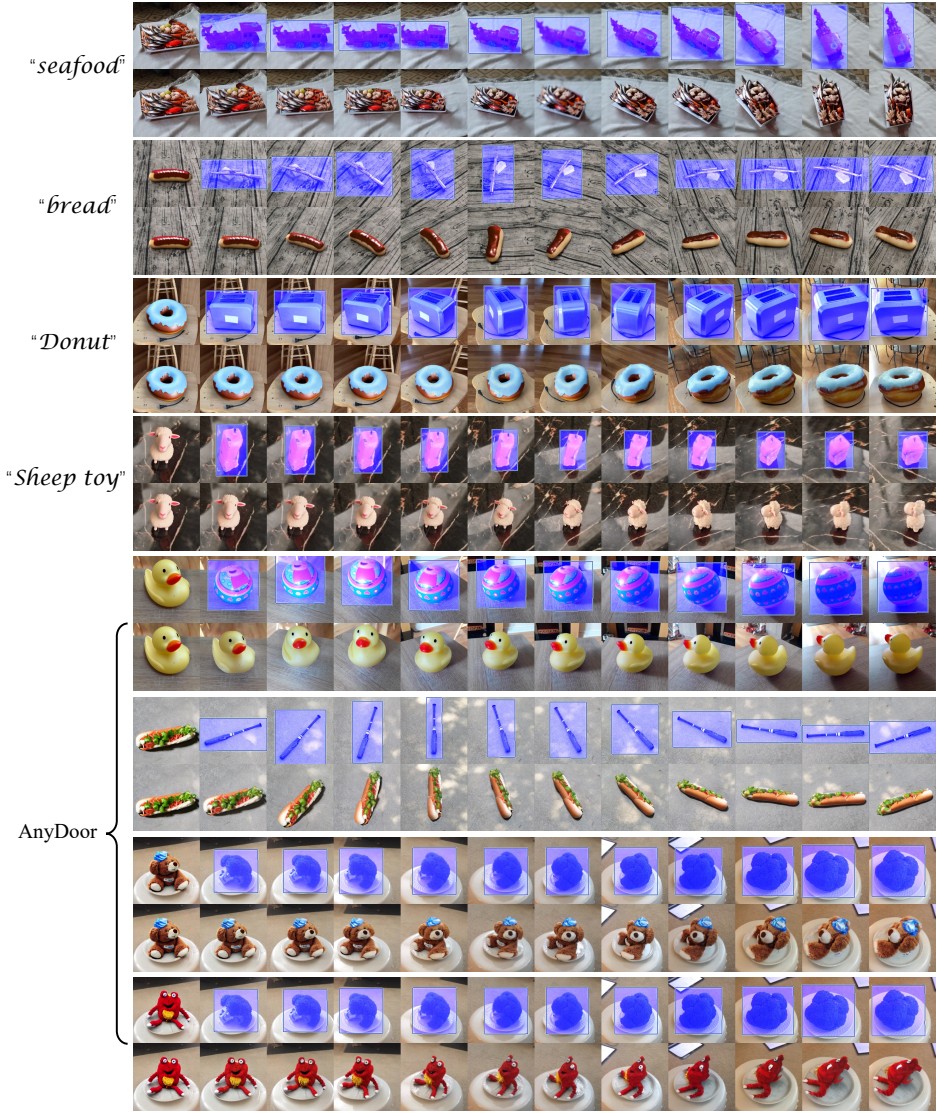

Figure 11: Object replacement results edited by T2I inpainting model and AnyDoor [14].

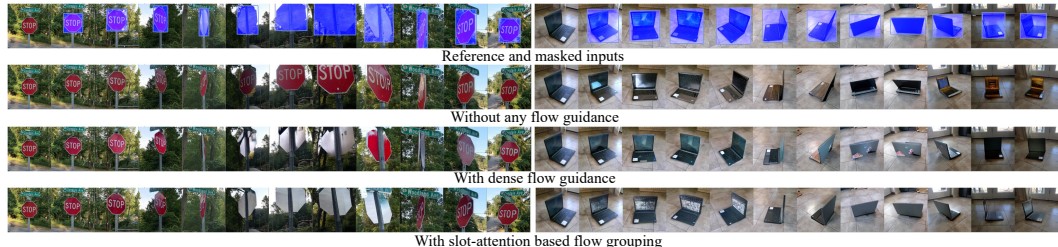

Figure 12: Qualitative ablation studies of flow guidance.

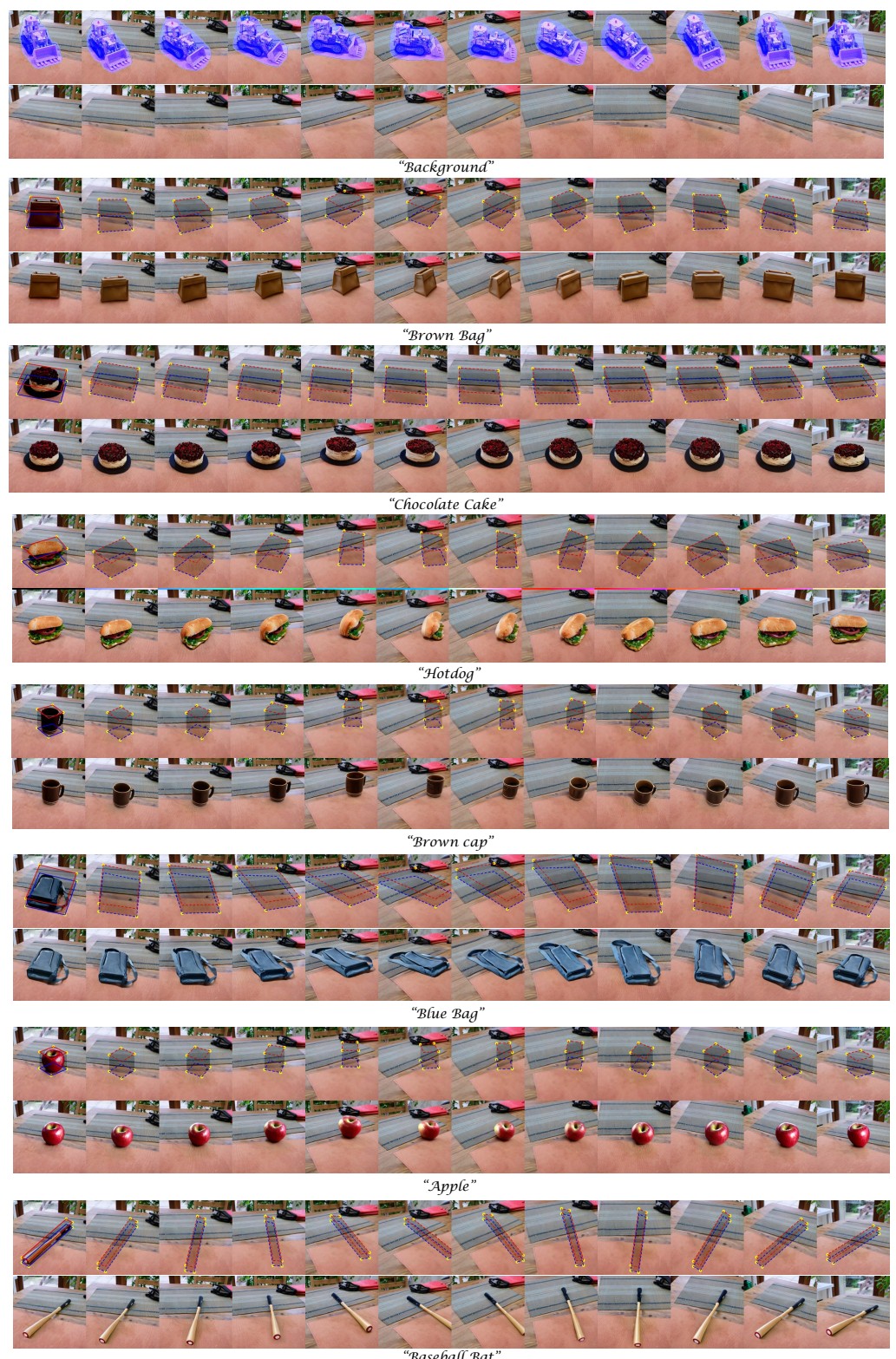

Figure 13: Scene editing results and adaptively warped masks with different captions. Object insertions are all based on the removal results with the caption: "background". Zoom-in for details.

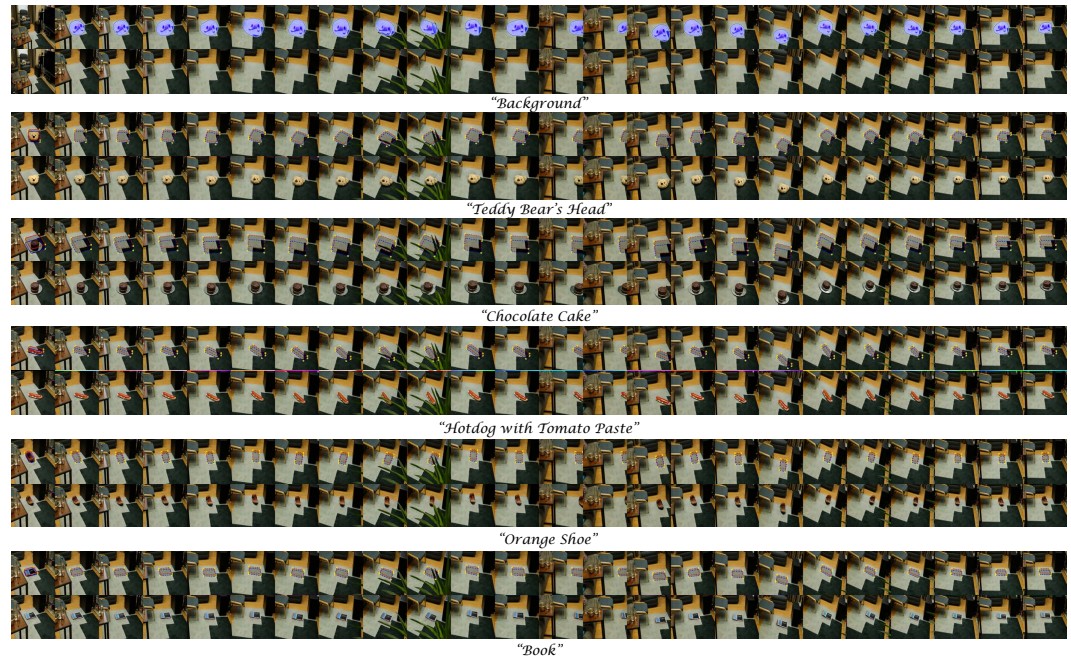

Figure 14: Scene editing results and adaptively warped masks with different captions. Object insertions are all based on the removal results with the caption: "background". Zoom-in for details.

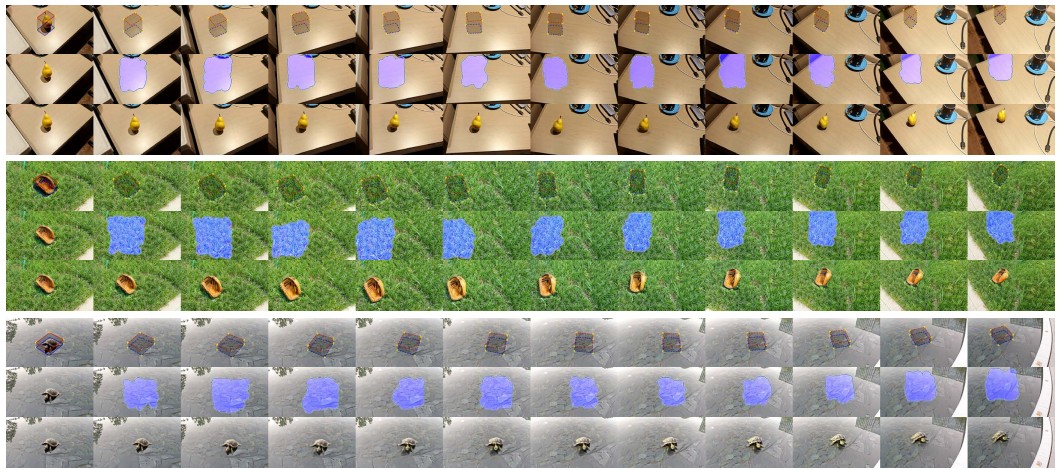

Figure 15: We show the robustness of the proposed mask adaption across various real-world scenarios, comprising the textureless table, the rough lawn, and the pool with sunlight reflection. SDXL-inpainting [54] is used to generate the object of the first view.

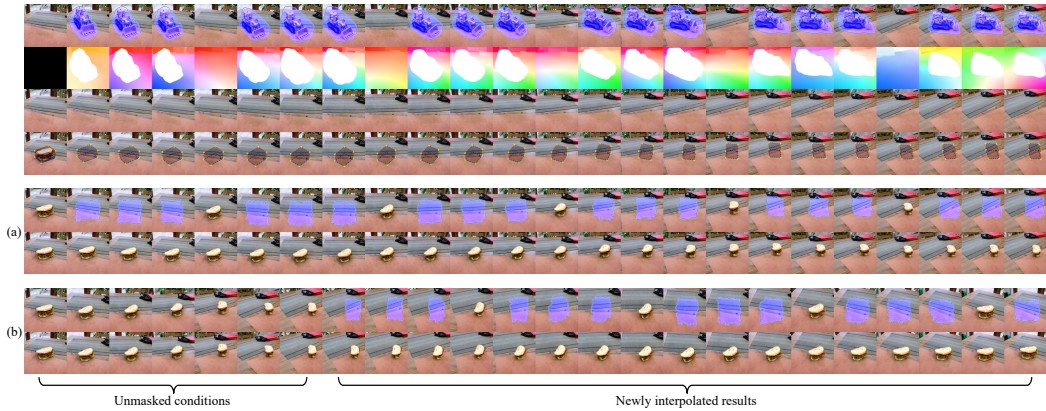

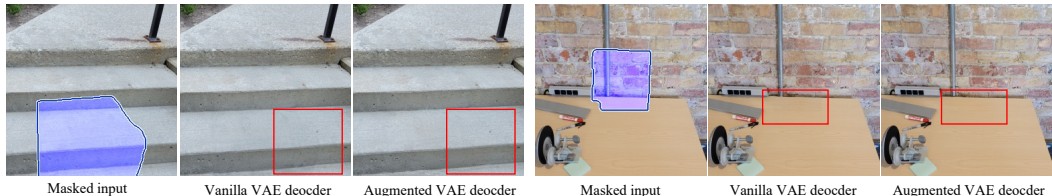

Unmasked conditions      Newly interpolated results

Figure 16: The visualization of frame interpolated object removal and insertion. (a) shows expanding results with (×4) length from 6 inpainted views. (b) denotes the long-range interpolation with fixed conditions (first 7 views).

Masked input    Vanilla VAE deocder    Augmented VAE deocder      Masked input    Vanilla VAE deocder    Augmented VAE deocder

Figure 17: Visualization of the color difference issue with different VAE decoders.

Table 5: Ablation study of the baseline method with inpainting formulation, and without inpainting formulation (SD-blend and SD-NVS).

| | PSNR↑ | LPIPS↓ | CLIP↑ |
|---|---|---|---|
| SD-blend | 14.35 | 0.443 | 0.648 |
| SD-NVS | 11.61 | 0.663 | 0.677 |
| Baseline | **17.16** | **0.305** | **0.750** |

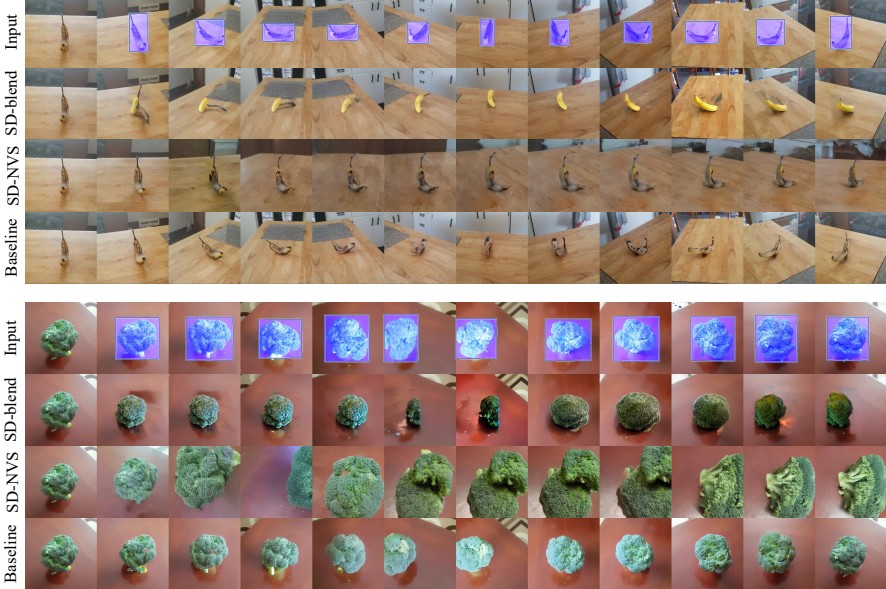

Figure 18: Results of baseline methods with (Baseline) and without inpainting formulation (SD-blend, SD-NVS).

inpainting fine-tuning and masking. The denoising process of 'SD-blend' is blended with unmasked regions [47], while 'Sd-NVS' is fine-tuned with the whole view synthesis with a noise-free first reference view. All models mentioned above are fine-tuned with random initialized LoRAs and motion transformer blocks without other improvements (Ref-KV, AnimateDiff, flow grouping) on CO3D. Both quantitative and qualitative results show the effectiveness of the inpainting-based baseline, which enjoys properly good pose formulation even without any pose guidance.

## B.5 Full Model Fine-tuning

Table 6: Comparison of MVInpainter-O based on fine-tuning only LoRAs and motion transformers with frozen backbone and the full-model fune-tuning. KID results are multiplied by 100.

| | CO3D+MVImgNet | | | | | Omni3D (zero-shot) | | | | |
|---|---|---|---|---|---|---|---|---|---|---|
| | PSNR↑ | LPIPS↓ | FID↓ | KID↓ | CLIP↑ | PSNR↑ | LPIPS↓ | FID↓ | KID↓ | CLIP↑ |
| LoRA+motion | 20.25 | 0.185 | 17.56 | 0.154 | 0.8182 | 19.19 | 0.153 | 16.40 | 0.345 | **0.7667** |
| Full fine-tuning | **20.76** | **0.181** | **17.51** | **0.134** | **0.8210** | **19.56** | **0.147** | **16.11** | **0.335** | 0.7633 |

We found that fine-tuning the whole model with 1e-5 learning rate for SD backbone on mixed CO3D and MVImgNet could further slightly facilitate the performance, as verified in Tab. 6. Thanks to the effective inpainting formulation, the LoRA and motion transformer based fine-tuning is visually good enough to handle our tasks. Moreover, training MVInpainter-O and MVInpainter-F with a shared SD backbone enjoys efficient and flexible real-world usage. Because only a few parameters would be resumed for different applications. We regard training a foundationally more powerful MVInpainter as interesting future work.

## B.6 Compared to NeRF Editing

We compare MVInpainter to the NeRF Editing method, SPIn-NeRF [51] in Fig. 19 and Tab. 7. Our contributions are orthogonal to NeRF editing-based manners [51]. MVInpainter focuses on tackling multi-view editing with a feed-forward model, while NeRF editing is devoted to reconstructing instance-level scenes with test-time optimization. NeRF editing manners require exact camera poses and costly test-time optimization for each instance (SPIn-NeRF needs about 1 hour for each scene). Moreover, NeRF editing manners fail to substitute for our method: a) NeRF editing starts with inconsistent 2D-inpainting results, which leads to blurred results as shown in Fig. 19, while our method could refer to a high-quality single-view reference without conflicts. b) Although both methods enjoy good consistency, rendering-based inpainting suffers from color difference when blended with the original images (the last row of Fig. 19). As shown in Tab. 7, our method is comparable to SPIn-NeRF in consistency (DINO-S, DINO-L) with better image quality (PSNR, LPIPS, FID) and fidelity in their official object removal test set.

Table 7: Object removal compared to SPIn-NeRF [51].

| | PSNR↑ | LPIPS↓ | FID↓ | DINO-S↑ | DINO-L↑ |
|---|---|---|---|---|---|
| Ours | **28.87** | **0.036** | **7.66** | **0.8972** | 0.5937 |
| SPIn-NeRF | 25.82 | 0.084 | 38.13 | 0.8681 | **0.6350** |

## B.7 Inference Time

Table 8: Inference time cost tested on A800 NVIDIA GPU. The view number is 24, while all inputs are resized into $256 \times 256$.

| Methods | Ours | AnimateDiff | Nerfiller | LeftRefill |
|---|---|---|---|---|
| DDIM steps | 50 | 50 | 20 | 50 |
| Time | 11.5s | 10.1s | 32.4s | 33.0s |

We validate the inference time of our MVInpainter, AnimateDiff [25], Nerfiller [80], and Leftrefill [7] in Tab. 8. Note that all methods are based on 50 steps of DDIM except for Nerfiller, which uses 20

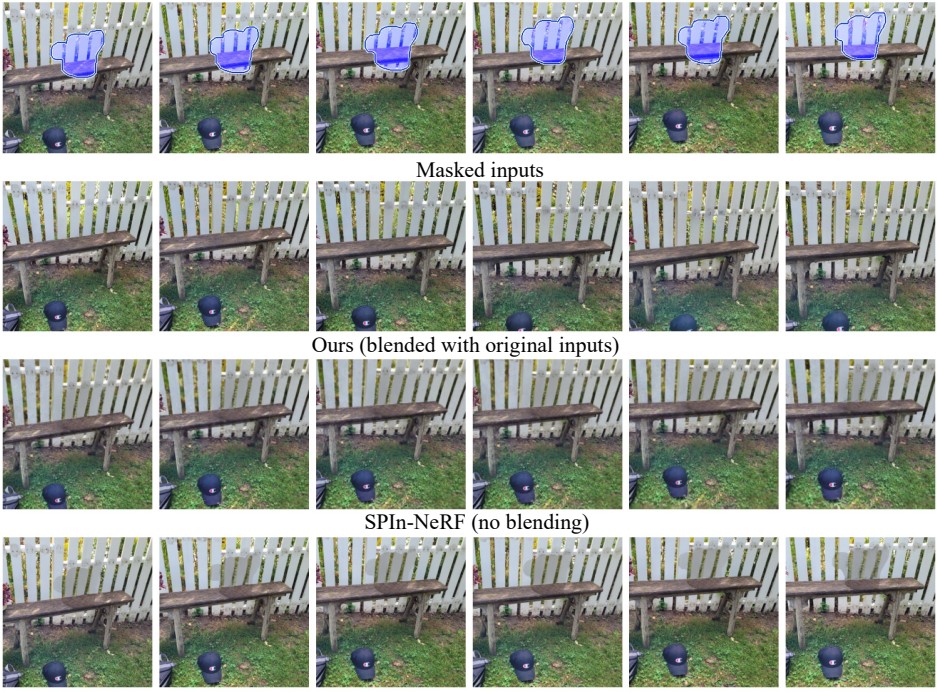

Masked inputs

Ours (blended with original inputs)

SPIn-NeRF (no blending)

SPIn-NeRF (blended with original inputs)

Figure 19: Object removal compared to SPIn-NeRF [51].

steps as the official setting. Our method is much more faster than other inpainting manners compared to Nerfiller and Leftrefill. Nerfiller requires costly iterative updating, which is slow even with much less de-noising steps. Leftrefill can only produce one view at once time rather than jointly produce all target views as ours. Our method only costs a little more inference time compared to the baseline AnimateDiff (+1.4s), which is mainly used by flow grouping and Ref-KV.

## C   3D Scene Reconstruction

### C.1   Dense Point Cloud

Since MVInpainter achieves consistent multi-view 2D generations, we simply employ some existing 3D reconstruction methods to further extend these results to 3D scenes without sophisticated pipelines.

**Dust3R.** First, we leverage the pose-free Dust3R [76] to achieve dense point clouds as the initialization of 3DGS. We empirically find that 12 views are sufficient to achieve high-quality point clouds as shown in Fig. 20(a) with only a few seconds. To ensure consistent initialization for 3DGS training, we align all point clouds from Dust3R with the same camera pose system of Colmap. Although the combined point clouds in Fig. 20(a) suffer a little inconsistency and outliers caused by errors from Dust3R estimation and alignment, 3DGS could eliminate them.

**Multi-View Stereo (MVS).** For some more difficult cases, such as 'room' in MipNeRF360, Dust3R fails to achieve consistent point clouds across all image pairs. Instead, we choose the SOTA learning-based MVS method, MVSFormer++ [9], to reconstruct the dense point cloud with 24 input views as shown in Fig. 21. MVS also takes a few seconds for the depth estimation and re-projecting correction, which is as efficient as Dust3R. The success of MVS verifies that MVInpainter can produce consistent multi-view inpainting, encouraging stereo-based reconstruction.

### C.2   3DGS Reconstruction

To train 3DGS, we first interpolate inpainted results of 'kitchen' in MipNeRF360 to 48 views with frame interpolation (Sec. B.2), where 42 frames are the training set; and 6 frames are the test set, while the initialized point clouds are got from Dust3R. Besides, we use only 24 inpainted views of 'room' to verify the robustness of our method with sparser input views in complicated scenes, while

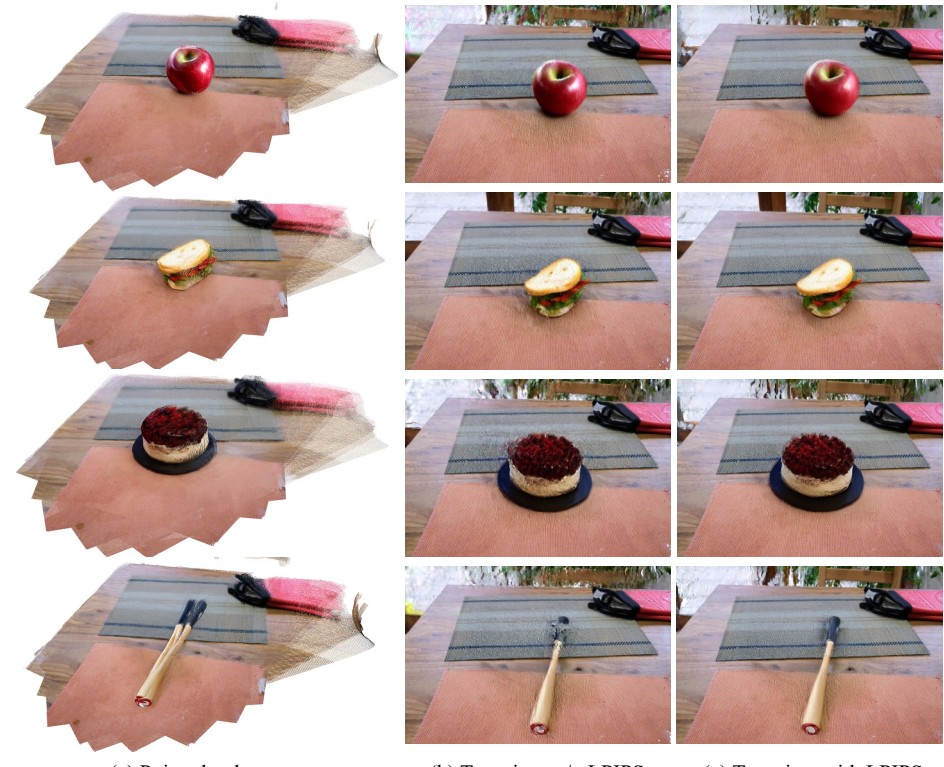

| (a) Point clouds | (b) Test view w/o LPIPS | (c) Test view with LPIPS |

Figure 20: The visualization of 3D scene reconstruction. (a) denote aligned point clouds produced by Dust3R [76]. (b) and (c) are test views of 3DGS with and without local LPIPS loss respectively. Dust3R fails to achieve consistent point clouds for the challenging baseball bat. But the local LPIPS loss can handle such a messily initialized 3DGS.

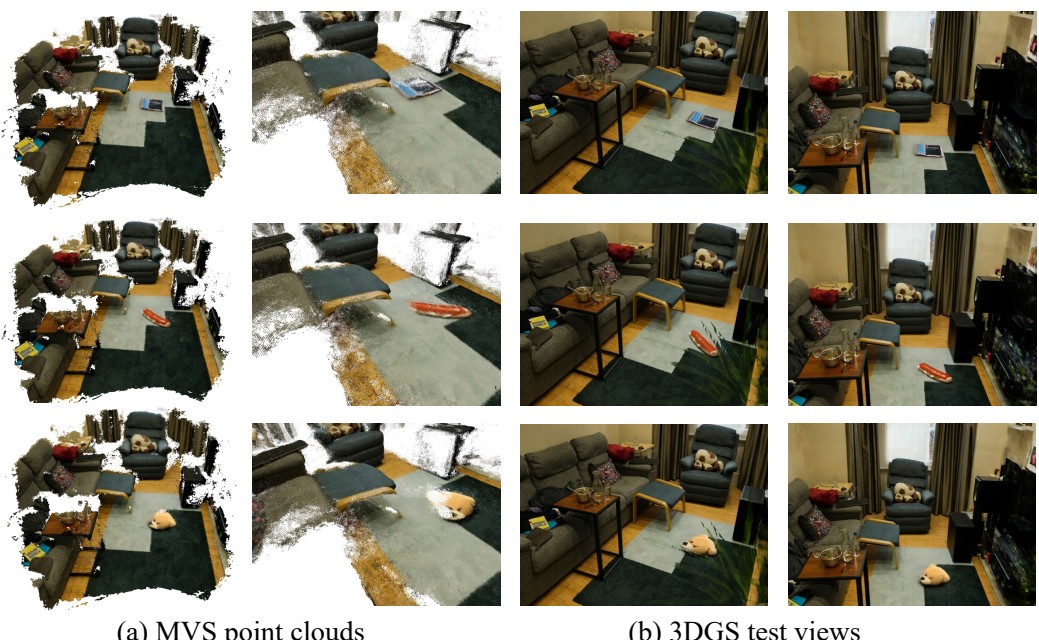

| (a) MVS point clouds | (b) 3DGS test views |

Figure 21: The visualization of 3D scene reconstruction based on MVS. (a) denote point clouds produced by MVSFormer++ [9]. (b) show rendered 3DGS test views.

MVSFormer++ is used to provide the initialized point clouds. Different from previous works that

require heavy SDS loss [64, 3] and dataset updates [26, 68, 80], the naive 3DGS could be simply converged with our raw multi-view results. Following [48], we find that LPIPS loss [99] could further alleviate the influence of slightly inconsistent appearance. However, learning 3DGS with only LPIPS and SSIM losses as [48] is very unstable for real-world scenes' reconstruction in our pilot study. Instead, we optimize LPIPS loss only for the foreground object as:

$$\mathcal{L}_{3dgs} = \lambda\mathcal{L}_1 + (1 - \lambda)\mathcal{L}_{ssim} + m \odot \lambda_{lpips}\mathcal{L}_{lpips}, \tag{4}$$

where $m$ is the foreground mask from SAM-tracking [88]; $\odot$ is element-wise multiplication; $\lambda = 0.2, \lambda_{lpips} = 0.1$ respectively. Thanks to the consistent multi-view results, we empirically find that just 5k training steps with 3 minutes could achieve good 3DGS results as in Fig. 20(b)(c). Furthermore, the local-based LPIPS loss further facilitates the performance with clear boundaries. Specifically, for the baseball bat in Fig. 20(a), the final dense point cloud struggles to perfectly unify point clouds from all views, because of the challenging long shape that magnified the Dust3R stereo and mask adaption errors. However, our local LPIPS-based 3DGS can largely alleviate this issue with consistent outcomes. We recommend comparing the video results of inpainted multi-view images and 3DGS in our supplementary.

## D  Limitations

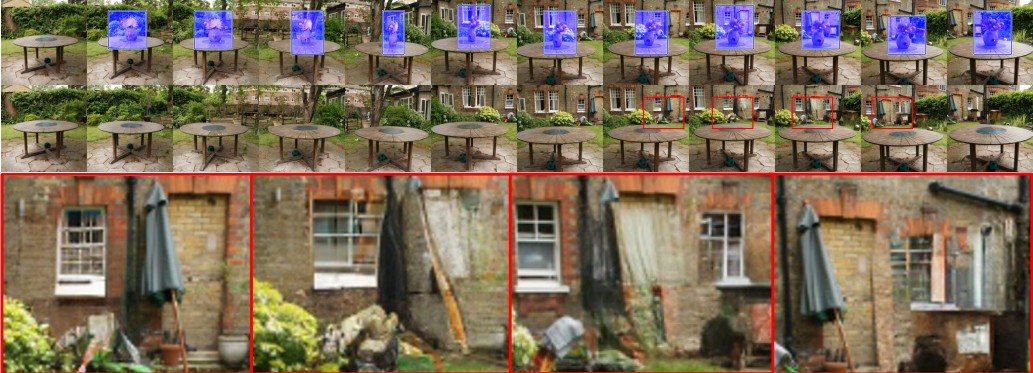

Figure 22: Limitations of MVInpainter-F, failing to tackle the complicated 360° scene inpainting, where the background is completely different from the reference one.

Although our work enjoys good consistency and outstanding generalization compared to previous manners, some dilemmas are still retained. As shown in Fig. 22, when meeting intractable 360° scene inpainting, our method achieves good results in the foreground regions which are consistently captured by the reference view. But our method fails to recover proper consistent structures for the completely unseen backgrounds, as magnified in red boxes of columns 8 and 9 in Fig. 22. This is caused by the limited respective fields of Ref-KV, and constrained training capacity (frozen SD backbone and limited training data of large viewpoint changes). However, we should clarify that the Ref-KV is sufficiently effective for most scenarios as verified in our ablation study, which is much more efficient than the full attention. Scaling up the multi-view inpainting model with more powerful attention mechanisms and more high-quality data is interesting future work.

## E  Broader Impacts

This paper exploited multi-view consistent inpainting based on text-to-image models. Because of their powerful generative capacity, these models would produce misinformation or fake images. So we sincerely remind users to pay attention to it. Besides, privacy and consent are important considerations, as generative models are often trained on large-scale data. Furthermore, generative models may perpetuate some biases according to the training data, leading to unfair outcomes. Therefore, we recommend users be responsible and inclusive while using these text-to-image generative models. Note that our method only focuses on technical aspects. Both images and pre-trained models used in this paper are all open-released.

