# OpenReview forum: "MVInpainter: Learning Multi-View Consistent Inpainting to Bridge 2D and 3D Editing"
_NeurIPS.cc/2024/Conference — NeurIPS 2024 poster_

### Official Review · Reviewer_NjjA · 2024-07-09

**Soundness:** 2
**Presentation:** 2
**Contribution:** 2
**Rating:** 4
**Confidence:** 3

**Summary:**

This paper proposes a new method for multi-view consistent inpainting. The input includes a video and a sequence of masks. The first image could be manipulated by any 2D editing approach, while the remaining images need to be inpainted using the proposed method. The method is built upon the SD1.5-inpainting model and fine-tuned with two LoRAs: one for object-centric and one for scene-level, using different datasets. It employs motion priors by fine-tuning the domain-adapted LoRA and temporal transformers from AnimateDiff, pre-trained on a video dataset. The technique of reference key & value concatenation from previous works is used to enhance consistency with the reference image. Flow features are extracted and added to the UNet as additional conditions. During inference, a heuristic strategy is proposed to warp and propagate the mask from the first frame to all subsequent frames.

**Strengths:**

1. The task of propagating a single 2D edited image to generate a consistent video is interesting.

2. The overall pipeline is well-motivated, built upon the SD1.5-inpainting base model, fine-tuning two LoRAs on different datasets, and including temporal modules, flow features, and reference attention techniques.

3. The proposed method doesn't require camera pose as input and has a short inference time.

4. The paper evaluates the method on several datasets with various tasks and also provides some ablation studies.

5. It's interesting to know that the dense feature is less effective than the slot embeddings.

**Weaknesses:**

1. While the title contains "to Bridge 2D and 3D Editing," the proposed techniques are primarily designed for 2D images or videos, with no explicit 3D representation used or 3D output generated. It would be interesting to explore whether 3D output can be extracted from the resulting image sequence. Otherwise, it might be better to change the title "3D editing" (and other mentions in the text) to something less misleading.

2. The utilization and fine-tuning of domain-adapted LoRA and temporal transformers from video models are intuitive and straightforward, but there do not appear to be any insightful design choices, limiting the novelty.

3. The technique of "Reference Key & Value Concatenation" has been used in many previous works. We fail to see much novelty here.

4. While the section introducing the techniques is generally clear, the concrete settings and details for training, inference, and evaluation are very hard to follow. I tried my best to understand the setup but still feel confused about many details, which is quite frustrating. The paper needs to be rewritten in a more well-organized way, introducing all necessary settings and details. Specifically, the following points need clarification:

    a) During inference, you employ 2D models to generate reference images for different tasks (e.g., object removal/editing/replacement). However, what is the input-output pair during training? For example, do you have specific ad-hoc training pairs for the object removal task?

    b) The experiment contains various setups, and the concrete settings are confusing. For each setup, it would be better to explicitly mention the task (e.g., inpainting with object/random masks, object generation, object removal, object replacement), dataset, the input, ground truth output, and how the **mask** are generated. Currently, it only says "Object-Centric Results," "Forward-Facing Results," and "Real-World 3D Scene Editing." But the concrete tasks are unclear. For example, what is the difference between "Object Removal" and "Scene-Level Inpainting"? I assume "object removal" is also achieved by inpainting? If the setup for quantitative and qualitative experiments is different, please also mention it. For example, there are some qualitative results for "object removal," but it should not be evaluated quantitatively (no ground truth)?

5. In equation (1), you concatenated both the masked latents and unmasked noised latents. Some related questions:

    a) The 9-channel input differs from the original input of SD1.5-inpainting. How do you achieve that?

    b) The noise-free latent is currently masked after VAE encoding. Will this cause leakage? It might be better to mask before VAE encoding.

6. What is the difference between the "Masking strategy" (line 163) and "masking adaptation" (line 228)? Is one for training and one for inference?

7. The method extracts flows before masking. This may be acceptable for the tasks of removal and replacement, but will it cause leakage for the task of general NVS and inpainting, considering the flow model may have seen the ground truth image?

**Questions:**

1. Have you considered directly fine-tuning a video inpainting model instead of using SD1.5-inpainting?

2. Line 186: The sentence "which always captures the unmasked reference latent without noise (Eq. 1), thus it is unnecessary to re-scale the latent before adding noise from another U-Net" is unclear to me. Could you please clarify?

3. Line 207: The sentence "except that the former should be normalized in the query dimension, while the latter is normalized in the key dimension" is unclear. Could you elaborate?

4. Line 212: The phrase "with comparable performance and fewer trainable weights" needs clarification. Which two setups are being compared?

5. Could you provide more details about the 3D attention mechanism?

6. Line 309: In the context of not having object-level tracking masks, how do you obtain the mask?

**Limitations:**

Yes, discussed in supplementary.

---

> ### Author Rebuttal · Authors · 2024-08-04
>
> We rearrange and classify similar questions together.
>
> **W1. No 3D representation? Tuning down "3D editing" in title.**
>
> Thanks.
> First, we would like to clarify that our paper has included a detailed discussion on the explicit 3D representation (3DGS) within our method in Section C of Appendix (Lines 727-762). Additionally, we have provided videos rendered using 3DGS in supplementary.
> Second, the primary contribution of our work is to 'bridge' the gap between 2D and 3D editing, rather than focusing exclusively on either 2D or 3D synthesis. This bridging is fundamentally about achieving multi-view consistent inpainting (Lines 45-46), which MVInpainter effectively addresses.
> Benefiting from the consistent inpainting, our method enjoys reliable 3D representation (Lines 298-299).
> While the key contribution of our work lies in multi-view inpainting, discussion on 3D representation holds a lower priority.
> Therefore, we provided them in the Appendix.
> We would further clarify this point in revision.
>
> **W2, W3. Limited novelty of LoRA, temporal transformers, and Ref-KV?**
>
> Thanks. While we understand your perspective, we respectfully disagree.
> The primary novelty of our paper lies not in the specific model design, but in the innovative task formulation that synergistically leverages video priors, optical flow, and the inpainting formulation to simplify the complex novel view synthesis (NVS) into a multi-view inpainting task (Lines 44-46).
> We believe that the novel model design is not the only way to advance AIGC development. An effective new formulation can significantly reduce task complexity (Lines 122-123), thereby also providing significant novelty and insights to the community.
> Besides, except for video inpainting modules, we also propose other novel and effective components, including flow grouping and mask adaption.
>
> **Q2. Clarifying for Line 186: "which always captures unmasked ... noise from another U-Net".**
>
> Thanks.  Line 186 details the differences between our usage of Ref-KV and previous works [Hu L et al,CVPR2024;Ruoxi Shi et al,2023]. We apologize for the incorrect citation here; AnyDoor[Chen, Xi, et al,CVPR2024] should be AnimateAnyone[Hu L et al,CVPR2024].
> Formally, AnimateAnyone used another diffusion U-Net to encode clear reference views (noise-free), while Zero123++ re-scaled the latent inputs by multiplying a constant to the reference latent (5x) to strengthen the effect of the reference view.
> In our work, no additional modules or adjustments are necessary to encode the reference latent. This is because the first reference frame in our inpainting model is completely mask-free, providing straightforward guidance when concatenated as the unmasked input (Lines 185-187).
> We will provide further clarification on these points and revise the citation.
>
> **W4. a) Setting details of tasks, datasets, and masks.**
> Thanks. We have introduced the inference pipeline in Section 3.4 and Figure 3(a) of the main paper. The overall tasks of this paper are clear, i.e., multi-view object removal (MVInpainter-F) and object insertion (NVS by MVInpainter-O), while the replacement is just the combination of these two tasks as in Figure 3(a).
> We introduced all input-output pairs for NVS and object removal in Lines 142-156, Figure 2(a) of the paper, which is further detailed in Table 4 of Appendix. NVS is trained on object-centric data (MVInpainter-O), while object removal is trained on face-forwarding data (MVInpainter-F). Masks are discussed in Lines 163-169.
>
> **Ad-hoc pairs for object removal?**
> No ad-hoc training pairs are needed for object removal.
> Based on the common conclusion of previous inpainting works, training on the scene-level data with random masks is sufficient to learn the object removal ability.
> We will further improve the task definition and data setting.
>
> **Q6. How to obtain training masks without object masks?**
> Without the object-level tracking masks, we only use hybrid random inpainting masks (Lines 163-164) to train MVInpainter-O.
> Note that we have discussed that both MVInpainter-O and MVInpainter-O adopt random inpainting masks, while we additionally employed object masks to MVInpainter-O (Lines 164-165).
>
> **b) Various setups of experiments, Difference between Object Removal and Scene-Level Inpainting.**
>
> Thanks. All sections are used to evaluate MVInpainter-O and MVInpainter-F separately.
> Formally, Section 4.1 (Object-Centric Results) is dedicated to evaluating the NVS performance of MVInpainter-O trained on object-centric data; Section 4.2 (Forward-Facing Results) is dedicated to evaluating the performance of MVInpainter-F trained on forward-facing data; Section 4.3 (Real-World 3D Scene Editing) shows the result of the combination of object removal (MVInpainter-F), NVS (MVInpainter-O) and mask adaption (Lines 296-298).
> We further divide Section 4.2 into 'Object Removal' and 'Scene-Level Inpainting' to evaluate the abilities to remove objects with object masks (Line 279) and inpaint images with random masks (Line 291). So 'Object Removal' and 'Scene-Level Inpainting' are both tested with MVInpainter-F but with different mask types and test sets.
> We also detailed the settings of datasets (Lines 246-254, Lines 675-687) and metrics (Lines 263-267) in the paper.
>
> **c) Setup for quantitative and qualitative experiments, especially for object removal.**
>
> Thanks. All quantitative and qualitative experiments are conducted on the same dataset except for the object removal (Line 279-280), which were discussed in paper.
> Quantitative results of object removal are based on the **test set** of SPInNeRF without foregrounds which can be regarded as GT (Line 279), while qualitative results of object removal are evaluated on the **train set** of SPIn-NeRF with foregrounds (no GT, Line 280). More details are discussed in Lines 681-684 of Appendix.
>
> **Limited by the rebuttal context, other concerns from reviewer NjjA are answered in the global rebuttal (at the top of this page).**

---

> > ### Author Response · Authors · 2024-08-12
> > **Feel free to discuss any remaining questions or concerns**
> >
> > Thanks for your valuable feedback. We have carefully addressed all your concerns and provided the details in our rebuttal. Our models and codes will be open-released, including all details to reproduce our results. As the rebuttal deadline approaches, please feel free to discuss any remaining questions or concerns. We will try our best to answer your questions.

---

### Official Review · Reviewer_n43z · 2024-07-12

**Soundness:** 3
**Presentation:** 3
**Contribution:** 3
**Rating:** 7
**Confidence:** 4

**Summary:**

The paper formulates the 3D object editing task as a multi-view 2D in-painting task. Firstly, MVInpainter-F is employed to remove the object and obtain the background scene. Then, MVInpainter-O generates multi-view images based on the reference view.

**Strengths:**

1. Solving the 3D object editing task as multi-view generation is interesting and innovative.
2. The thoughtful selection of different training datasets (indoor scene/object-centric) for training MVInpainter-F and MVInpainter-O effectively decomposes the object replacement task.
3. The extensive experiments, including the generation of long sequences and the 3D scene reconstruction using the generated multi-view images, further demonstrate the 3D consistency of the resulting outputs.

**Weaknesses:**

It is uncertain whether the design of mask adaption is robust enough.

**Questions:**

1. During training, the flow is obtained using RAFT and masked. In the inference stage of MVInpainter-F, is the flow only obtained from the masked image? Does the utilization of this low-quality flow negatively impact the results?
2. How can we ensure that mask adaption remains robust when dealing with different masks and backgrounds?

**Limitations:**

The author provides a detailed discussion of limitations and broader impacts in the supplementary material.

---

> ### Author Rebuttal · Authors · 2024-08-04
>
> **W1, Q2. Whether mask adaption is enough robust? How to deal with different masks and backgrounds?**
>
> Thanks for this good point. As long as the 'basic plane' where the object is placed can be approximated to a plane, the proposed mask adaption is robust enough with the theoretical basis of perspective transformation. We have shown the effectiveness of the mask adaption with various object shapes in Figures 12 and 13 of the Appendix.
> To further clarify the robustness, we provide more examples with complicated backgrounds in the wild shown in Figure 4 (rebuttal pdf), including textureless table, rough lawn, and pool with sunlight reflection.
> We find that the dense matching (RoMa[1]) used in our work enjoys good generalization on various backgrounds. Importantly, most backgrounds can be approximated to the 'basic plane' as claimed in Lines 231-233.
> We would add these results to the paper.
>
> [1] Johan Edstedt, et al. RoMa: Robust Dense Feature Matching. CVPR2024.
>
> **Q1. Do low-quality flows obtained from masked images negatively impact the result?**
>
> Thanks. We have discussed this point in the Line 194 (footnote of page 5).
> We extract flows before masking because foregrounds largely benefit the flow quality. Note that we process flows with 5-pixel dilated masks to avoid any leakage.
> Given that optical flows are typically low-level local features, we have not observed any conflicts when using masked flows as guidance. Furthermore, the quantitative results of dense flow presented in Table 3(b) verify that no leakage occurs (dense flows did not improve the NVS results), ensuring the integrity of quantitative results.
> We would clarify about the detailed operation of flow masking (mask dilation) in the revision.

---

### Official Review · Reviewer_VJEq · 2024-07-14

**Soundness:** 2
**Presentation:** 3
**Contribution:** 3
**Rating:** 5
**Confidence:** 3

**Summary:**

This paper proposes a new 3D editing method by regarding it as a multi-view 2D inpainting task.  It ensures cross-view consistency through video priors and concatenated reference key/value attention and controls camera movement without explicit poses using slot attention. It shows effectiveness in object removal, insertion, and replacement.

**Strengths:**

- The slot-attention-based flow grouping has a novel and reasonable design. As demonstrated in various video/multi-view models, self-attention-based key and value concatenation is effective in ensuring view consistency. Overall, this paper proposes a well-structured recipe for multi-view editing.

- The strategies used in the proposed method and their justifications are well explained in detail.

- The video results and quantitative results demonstrate good performance.

**Weaknesses:**

- Some assumptions are heuristic, such as in L232. While these assumptions may be reasonable in most cases, they might not be robust across diverse user cases.
- Overall, the figures and flow of the paper are complicated. It would be beneficial if the content of the figures were made more comprehensive.

**Questions:**

How is the generalizability? What is the performance in completely OOD cases?

**Limitations:**

the authors adequately addressed the limitations.

---

> ### Author Rebuttal · Authors · 2024-08-04
>
> **W1. Heuristic assumption of mask adaption?**
>
> Thanks for this point. We should clarify that the planar assumption of mask adaption in Line 232 is reasonable and straightforward to implement across diverse real-world cases.
> First, this strategy roughly decides the mask location rather than providing exact mask shapes, which would be further irregularly masked for better generalization (Line 241-242).
> Second, the mask adaptation proves to be robust for the 'bottom face' of targets with various shapes, as verified in Figures 12 and 13 of the Appendix. Almost all objects have explicit or approximate 'bottom faces' which makes this assumption widely applicable.
> For the `basic plane', the key ability is the dense matching method.
> We find that the dense matching (RoMa [1]) used in our work can be generalized to various wild backgrounds (textureless table, rough lawn, and pool with sunlight reflection) as verified in Figure 4 (rebuttal pdf), denoting the robustness of mask adaption.
>
> [1] Johan Edstedt, et al. RoMa: Robust Dense Feature Matching. CVPR2024.
>
> **W2. Complicated figures and flow? Needing comprehensive content of figures.**
>
> Thanks for this good advice!
> Indeed, finishing a 3D editing is an inherently non-trivial task, including sophisticated pipelines as seen in many pioneering works.
> In this paper, we simplify the overall pipeline into the following steps:
>
> (1) 2D inpainting → (2) multi-view removal (optional) → (3) multi-view synthesis/insertion → (4) 3D reconstruction (optional).
>
> These steps are illustrated in Figure 3 and detailed in the inference pipeline in Section 3.4.
> Our work focuses on steps (2) and (3), which are addressed by MVInpainter-F and MVInpainter-O respectively (Figure 2a).
> We agree with the reviewer that providing a more comprehensive overview figure between Figures 2 and 3 could make readers understand our paper better. We design a new overview shown in Figure 5 (rebuttal pdf), which clearly indicates the steps that are the main focus of our paper and identify the models specifically designed for each step.
> We will re-organize these points, if accepted.
>
> **Q1. Generalizability for OOD cases.**
>
> Thanks. Benefiting from the T2I prior from StableDiffusion, our method enjoys the capacity to tackle OOD cases, which have been verified in our submission.
> Formally, 15 categories of MVImgNet are unseen test sets (Line 678). We further evaluate the OOD capability of MVInpainter in the zero-shot Omni3D (Table 1, Figures 4 and 9 of the paper).
> We show some qualitative NVS results of OOD categories in Figure 3 (rebuttal pdf).
> Moreover, our method could also be used for unseen toys generated from exemplar-based synthesis[2] as verified in Figure 10.
> Considering the difficulty of scene-level NVS, it is challenging to properly address some completely OOD cases, such as human bodies, which are limited by the capacity of SD and AnimateDiff, as well as the restricted in-the-wild multi-view datasets (CO3Dv2+MVImgNet).
> Despite these limitations, MVInpainter is, to the best of our knowledge, the most robust reference image-guided method that generalizes well to in-the-wild object NVS across various categories (Lines 253-254).
> We would discuss this in the revision, and consider scaling up our model with more diverse training data to develop a foundational model as interesting future work.
>
> [2] Xi Chen, et al. Anydoor: Zero-shot object-level image customization. CVPR2024.

---

### Official Review · Reviewer_jssh · 2024-07-15

**Soundness:** 3
**Presentation:** 3
**Contribution:** 3
**Rating:** 6
**Confidence:** 3

**Summary:**

The paper studies the task of multi-view consistent real-world object removal and insertion, enabled by learning a model, MVInpainter, trained to perform multi-view 2D inpainting. The paper demonstrates its effectiveness on both object-centric and scene-level datasets with the task of object removal and object insertion.

**Strengths:**

The task that the paper addresses is important yet less explored. Learning strong priors for scene level multi-view consistent editing can serve as a basis model for various 3D applications.

The method introduces motion priors and optical flow to guide the generation, which seems to be interesting and effective with various quantitative and qualitative evaluations.

The presentation is good and the paper is overall easy to read.

**Weaknesses:**

1.Though the paper tackles the setting of video input (consecutive frames to obtain the motion prior), I am curious how the method performs with just multi-view inputs? Does the performance significantly drop? If the model can be applied to those as well, it could potentially lead to bigger impact and be applied to more real-world scenarios.

2.How are the priors different from AnimateDiff and Flow Grouping? Why are they complementary?

3.Could the authors please explain why the proposed method cannot be compared to SPIn-NeRF-type methods? Is it because of camera poses? If so, I am still curious to see how large the gap is? If the method is close to the methods with camera poses, it would make the paper stronger.

4.One limitation is that the method still works on images with a relative simple background, as pictured in the supplementary materials Figure 19.

**Questions:**

I think the paper is overall of good quality and tackles an important task. Some questions (as listed) could be explained for a better understanding of the method.

---

> ### Author Rebuttal · Authors · 2024-08-04
>
> **W1. How does the method perform with unordered multi-view inputs?**
>
> Thanks. Although our work mainly focuses on the sequential multi-view inputs to better leverage the video prior from video components, it can also achieve comparable performance with unordered inputs as exploratively verified in Table1 and Table2 (rebuttal pdf).
> Unordered inputs still enjoy competitive FID, DINO, and CLIP scores for object-centric novel view synthesis (NVS) and object removal, indicating good image quality and consistency. The proposed flow grouping and Ref-KV also strengthen the generated direction and appearance for unordered inputs, preserving stable PSNR. However, unordered NVS suffers from some structural distortions caused by irregular viewpoint changes as shown in Figure 1 of the rebuttal pdf. Thus, we would still clarify that using ordered images with video prior is a more efficient way to model the multi-view inpainting (Line 171-173). Maybe further fine-tuning our model with unordered images could alleviate this issue, which is beyond the scope of this paper.
>
> **Table1: Unordered generative results of object removal**
>
> |                       | PSNR ↑ | LPIPS ↓ | FID ↓ | DINO-S ↑ | DINO-L ↑ |
> |-----------------------|--------|---------|-------|----------|----------|
> | Ours (ordered)    | **28.87** | **0.036** | **7.66** | **0.8972** | **0.5937** |
> | Ours (unordered)      | 28.57  | 0.039   | 9.76  | 0.8943   | 0.5752   |
>
> **Table2: Unordered generative results of object-centric NVS**
>
> |                       | PSNR ↑ | LPIPS ↓ | FID ↓ | CLIP ↑   |
> |-----------------------|--------|---------|-------|----------|
> | Ours (ordered)    | **20.25** | **0.185** | 17.56 | **0.8182** |
> | Ours (unordered)      | 19.29  | 0.233   | **17.37** | 0.8173   |
>
> **W2. How are priors different from AnimateDiff and Flow Grouping? Why are they complementary?**
>
> Thanks. The priors from AnimateDiff and flow grouping serve different but complementary purposes in our method. The video prior from AnimateDiff enhances structure consistency in the generated outputs (Line 307-308 and Figure 6 of the paper). This helps maintain a coherent structure across frames. Flow Grouping improves pose controllability (Line 191-193 and shown in Figure 11 of the Appendix). This component ensures that the synthesized poses are accurate and aligned with the intended viewpoint changes from unmasked regions. For the NVS inpainting, objects are consistently masked, leading to potential ambiguities in some scenes. These ambiguities can cause errors in pose generation, but video models alone struggle to address them (Figure 11). Quantitative results further verify this point (Table 3).
>
> **W3. Why not compare to SPIn-NeRF-type methods? Because of the requirement of camera?**
>
> Thanks for this good point. Our contributions are orthogonal to NeRF editing-based manners (SPIn-NeRF[1]).
> MVInpainter focuses on tackling multi-view editing with a feed-forward model, while NeRF editing is devoted to reconstructing instance-level scenes with test-time optimization. Except for the exact camera poses, NeRF editing manners require costly test-time optimization (Line 109-112) for each instance (SPIn-NeRF needs about 1 hour for each scene).
> Moreover, NeRF editing manners fail to substitute for our method as evaluated in Figure 2 and Table 3 (rebuttal pdf).
> a) NeRF editing starts with inconsistent 2D-inpainting results, which leads to blurred results as shown in Figure 2 (rebuttal pdf), while our method could refer to a high-quality single-view reference without conflicts.
> b) Although both methods enjoy good consistency, rendering-based inpainting suffers from color difference when blended with the original images (the last row of Figure 2 (rebuttal pdf)).
> As shown in Table 3 (rebuttal pdf), our method is comparable to SPIn-NeRF in consistency (DINO-S, DINO-L) with better image quality (PSNR, LPIPS, FID) and fidelity in their official object removal test set.
> Note that our method is orthogonal to instance-level manners and can further boost their performance with much better inpainting initialization.
> For example, experiments in Section C of the Appendix show that our method could be easily integrated with 3DGS to achieve consistent 3D outputs.
> We would add these discussions to the paper, if accepted.
>
> **Table3: Object removal compared to SPIn-NeRF**
>
> |             | PSNR ↑    | LPIPS ↓  | FID ↓    | DINO-S ↑    | DINO-L ↑    |
> |-------------|-----------|----------|----------|-------------|-------------|
> | Ours    | **28.87** | **0.036**| **7.66** | **0.8972**  | 0.5937      |
> | SPInNeRF| 25.82     | 0.084    | 38.13    | 0.8681      | **0.6350**  |
>
> [1] Ashkan Mirzaei, et al. Spin-nerf: Multiview segmentation and perceptual inpainting with neural radiance fields, CVPR2023.
>
> **W4. Limitation of complex background.**
>
> Thanks. It is important to note that all methods have certain constraints. Our approach is currently limited by the capabilities of StableDiffusion and AnimateDiff, making it challenging to inpaint 360° complex backgrounds that are out of reference view.
> However, our method is effective enough for most multi-view scene editing. We acknowledge that scaling up our model to a foundational video inpainting model would be interesting future work.

---

### Author Rebuttal · Authors · 2024-08-04

We appreciate the valuable comments from all reviewers. We thank the positive comments of 'interesting and effective', 'novel and reasonable design' of slot-attention based flow grouping (jssh, VJEq); 'important yet less explored', 'interesting and innovative' of multi-view generation (n43z, NjjA). We address remained concerns for all reviewers. While our experiments are solid enough, we further provide some supplementary results in the rebuttal PDF to make our claims more convincing. Please refer to the rebuttal PDF for more details and results.

Here we further address remaining concerns from reviewer NjjA.

**W5. Questions about inpainting form (Eq1).
a) 9-channel input differs from SD1.5-inpainting?**
Thanks. Our model is based on the SD1.5-inpainting (Lines 124-125), which originally contains 9-ch input, including noised latent (4-ch), mask (1-ch), and masked latent (4-ch).

**b) Why is noise-free latent masked after VAE encoding?**
Thanks for careful reading. Sorry for the confusing presentation of Eq(1). The masking is done before VAE encoding. Here we just want to denote the masked latent. We would revise Eq(1) in the revision.

**W6. Difference between 'masking strategy' (line 163) and 'masking adaption' (line 228)?**
Thanks. Yes, the masking strategy in Line 163 is introduced for training, while the masking adaptation in Line 228 is introduced for inference with good generalization as verified in Figures 12, 13, and Figure 4 (rebuttal pdf).

**W7. Leakage of flows in NVS?**

Thanks for this good point.
We clarify that the flow used in our work does not leak information for NVS and inpainting:
1) As in the footnote of page 5, our method involves extracting flows from unmasked images and then applying masks to them. To further prevent leakage, we dilate flow masks by 5 pixels.
Given that optical flows are typically low-level local features, it is challenging for them to carry masked clues to unmasked regions. No conflicts are observed when we use masked flows as guidance for removal and insertion.
2) The proposed method leverages flow grouping with slot-attention to extract high-level motion from flow features (Line 200). These features primarily capture rough pose directions in unmasked regions rather than detailed information, making it difficult to leak masked info.
3) Importantly, quantitative results of dense flow injection in Table 3b indicate that simply adding flow features does not improve NVS quality. If there were any potential leakages in these masked flows, NVS results would be rapidly improved, which is not observed.

We will include detailed explanations of flow masking operations, including mask dilation in the revision.

**Q1. Fine-tuning a video inpainting model instead of SD1.5-inpainting?**

Thanks. We agree that fine-tuning a foundational video inpainting model is a promising direction to strengthen our work, which could be regarded as future work.
Unfortunately, to the best of our knowledge, there are currently no open-released video inpainting models with sufficient capacity to address the NVS task mentioned in this paper. Existing video inpainting models fail to tackle our tasks (Lines 82-83).
On the other hand, fine-tuning a foundational video like SVD into a video inpainting model requires substantial computational resources and data, which is beyond the scope of this paper.
However, our approach, which unifies 2D-inpainting and AnimateDiff, proves both efficient and effective with good convergence (Lines 175-176).
Moreover, we believe that once a foundational video inpainting model becomes publicly available, MVInpainter can be seamlessly integrated into this new model, potentially yielding better performance.

**Q2 is answered in the previous rebuttal page.**

**Q3. Why slot-attention is normalized in query dimension?**

Thanks. Line 207 clarifies the difference between slot attention[1] and vanilla cross-attention.
Given $K$ query (slot) features $\mathbf{Q}\in\mathbb{R}^{K\times d}$ and key features $\mathbf{K}\in\mathbb{R}^{HW\times d}$ with different length $K$ and $HW$, we can achieve the attention matrix: $\mathbf{Q}\mathbf{K}^T\in\mathbb{R}^{K\times HW}$.
Following official implementation of slot attention[1], the softmax normalization is applied along the attention matrix over $K$ slot dimension. This ensures that attention coefficients sum to one for each slot query. This is why we mentioned that slot attention (called 'former' of Line 207) normalizes in the query dimension.
In contrast, the vanilla cross-attention (called 'latter' of Line 207) normalizes along the $HW$ key dimension.
According to [1], such normalization in the slot dimension improves the stability. We recommend referring to [1] for more details.
We will clarify this point in the revision.

[1] Object-centric learning with slot attention. NeurIPS2020.

**Q4. Line 212: Which two setups are compared?**

Thanks. We have compared different ways to inject flow grouping into our model in Table 3b.
The way 'cross-attn' (Line 212) performs similar to 'time-emb' (injecting features through trainable AdaNorm before conv and attn), while the latter needs to train more weights for AdaNorm layers. We will revise it to 'ada-norm-emb' for a clearer presentation.

**Q5. More details about 3D attention mechanism.**

Thanks. The 3D attention mechanism refers to 'temporal 3D attention for the flow grouping' in Line 213.
In Figure 2(c), each flow grouping block contains a spatial-attention layer and a temporal-attention layer. This design is similar to the interleaved spatial-temporal attention layers used in video models. Since masked flow features contain rich temporal information, it is intuitive to extend slot-attention into 3D (spatial-temporal) receptive fields, making slot features carry more general information across all sequential views (Line 213-215). Quantitative results (Table 3b) also verified this. We will further clarify this.

**Q6 is answered in the previous rebuttal page.**

---

> ### Author Response · Authors · 2024-08-12
> **Feel free to discuss any remaining questions or concerns**
>
> Thanks for the valuable feedback from all reviewers. We have carefully addressed all concerns in our rebuttal. Our models and codes will be open-released, including all details to reproduce our results. As the rebuttal deadline approaches, please feel free to discuss any remaining questions or concerns. We will try our best to answer your questions.

---

### Decision · Program_Chairs · 2024-09-25

**Decision:**

Accept (poster)

**Comment:**

All reviewers acknowledged the novelty of this paper, as well as the solid and thorough experiments conducted. In the initial reviews, there were suggestions to further demonstrate the superiority of the proposed method from various perspectives. Especially, one reviewer raised several questions, including about explicit 3D representations based on our multi-view synthesis and technical details. During rebuttal, the authors tried to clarify the contribution of this paper and technical details. AC has also carefully reviewed the paper, the reviews, and the rebuttal, and concurs with most reviewers' opinions, such as the innovative conceptual contribution. Although some technical details could lead the misleading, AC thinks that these could be addressed in the final version. Therefore, AC recommends acceptance. It would be greatly beneficial to include the additional results from the rebuttal phase in the final version.